# Precise tumor immune rewiring via synthetic CRISPRa circuits gated by concurrent gain/loss of transcription factors

Yafeng Wang [1,2,7], Guiquan Zhang[1,7], Qingzhou Meng[3,7], Shisheng Huang[4], Panpan Guo[2], Qibin Leng[3], Lingyun Sun[2], Geng Liu [1,5✉], Xingxu Huang [4,6✉] & Jianghuai Liu [1,5✉]

Reinvigoration of antitumor immunity has recently become the central theme for the development of cancer therapies. Nevertheless, the precise delivery of immunotherapeutic activities to the tumors remains challenging. Here, we explore a synthetic gene circuit-based strategy for specific tumor identification, and for subsequently engaging immune activation. By design, these circuits are assembled from two interactive modules, i.e., an oncogenic TF-driven CRISPRa effector, and a corresponding p53-inducible off-switch (NOT gate), which jointly execute an AND-NOT logic for accurate tumor targeting. In particular, two forms of the NOT gate are developed, via the use of an inhibitory sgRNA or an anti-CRISPR protein, with the second form showing a superior performance in gating CRISPRa by p53 loss. Functionally, the optimized AND-NOT logic circuit can empower a highly specific and effective tumor recognition/immune rewiring axis, leading to therapeutic effects in vivo. Taken together, our work presents an adaptable strategy for the development of precisely delivered immunotherapy.

[1] State Key Laboratory of Pharmaceutical Biotechnology, Model Animal Research Center at Medical School of Nanjing University, Nanjing 210061, China. [2] Department of Rheumatology and Immunology, The Affiliated Drum Tower Hospital of Nanjing University Medical School, Nanjing 210008, China. [3] Affiliated Cancer Hospital & Institute of Guangzhou Medical University, 78 Hengzhigang Road, Guangzhou 510095, China. [4] School of Life Science and Technology, ShanghaiTech University, Shanghai 201210, China. [5] Jiangsu Key Laboratory of Molecular Medicine, Medical School of Nanjing University, Nanjing 210093, China. [6] Zhejiang Laboratory, Hangzhou 311100, China. [7] These authors contributed equally: Yafeng Wang, Guiquan Zhang, Qingzhou Meng. ✉email: liug53@nju.edu.cn; huangxx@shanghaitech.edu.cn; liujianghuai@nju.edu.cn

The recent clinical successes in cancer immunotherapy have inspired active advances in this discipline[1,2]. Given that most tumors are poorly immunogenic and reside within tolerant microenvironments, further developments of tumor tissue-localized (instead of systemic) immunological interventions are strategically desirable[3]. Such approaches may potentially drive clinical responses with favorable efficacy/toxicity profiles, and subsequently, help to address the current unmet medical needs in cancer treatments.

Latest developments in synthetic gene circuits have suggested them as promising tools for tumor-specific delivery of immunotherapeutic reagents[4–6]. Since dysregulation in transcription factor (TF) activities represent key malignant determinants[7], many synthetic gene circuits have been designed to sense the oncogenic TF (onco-TF) activities in tumors and to conditionally drive diverse therapeutic outputs[8,9]. Strategies of combinatorial sensing of more than one onco-TFs (AND logic) have also been implemented to improve tumor-targeting specificity[10–12]. However, as the onco-TFs often exert important functions in normal cells when properly regulated[13], their activities are inadequate to stringently distinguish tumorous and normal cells. In contrast, loss of tumor-suppressive TF activities is highly characteristic of tumors. p53 is a prominent example in this regard, being well-known as "guardian of the genome" and indeed the most frequently mutated gene in human cancers[14–17]. Moreover, even in tumors retaining the WT p53, it is believed that alterations in the related regulatory pathways often lead to the inactivation of p53 protein function[18,19]. Such strong prevalence of p53 functional deficiency in cancers offered a unique possibility of incorporating sensors of p53 status in gene circuits to empower highly accurate tumor identification and subsequent output production, e.g., to engage immune rewiring only upon sensing the co-occurrence of an onco-TF activation and p53 deficiency (AND–NOT logic). Besides an obvious advantage in specific recognition of tumor cells, a gene circuit with an AND–NOT logic is more capable of driving a robust output than an AND circuit, as any constraint established by the NOT gate would become fully removed under the targeted condition (e.g., p53-inactivation).

From a therapeutic point of view, it is highly desirable for synthetic gene circuits to incorporate a programmable actuator[5,20,21]. The CRISPR/dCas9-based transcriptional effectors (CRISPRa/i) have rapidly taken the center stage as powerful devices to program transcriptional activation or repression of endogenous genes, including those involved in immune regulation[22,23]. Moreover, such a versatile and modular synthetic tool is adaptable for information processing in gene circuits[21]. Interestingly, some recent works have also demonstrated another dimension of harnessing the natural anti-CRISPR proteins to exert effective control over various CRISPR devices[24–26]. These advances have made CRISPRa/i effectors an attractive platform for the construction of tumor-specific immune-rewiring gene circuits.

In the present study, we adopt various promoters and the dCas9 effector toolkit for the construction of synthetic gene circuits capable of identifying tumor cells by their combinatorial gain and loss of key TF activities (AND–NOT logic) (Fig. 1a and Supplementary Fig. 1a). Upon actuation, such a synthetic device empowers highly specific, tumor-activated immune stimulation and exhibits therapeutic effects in vivo. Our study establishes a promising platform for the development of precisely delivered anti-tumor immune interventions.

## Results

### An active CRISPRa can flexibly program immunostimulatory outputs in tumor cells.
Synthetic gene circuits hold considerable potential to enable tumor-specific delivery of immunological interventions[12]. We reasoned that the recently established CRISPRa effector platform might be further adapted as key parts for the construction of tumor-targeting immunoregulatory gene circuits. A potent, three-component synergistic activation mediator (SAM) effector was chosen for exploration[27,28]. This CRISPRa unit standardly consists of dCas9-VP64 (or unmodified dCas9), targeting sgRNA for activation (sgTGTa), and an artificial co-activator (MS2–p65–HSF1, MPH) specifically recruited by the engineered stem–loops on sgTGTa (Supplementary Fig. 1b). As the fused VP64 moiety on dCas9 was previously shown to be unessential for SAM-mediated gene activation[27], the "basic" version of dCas9 was used for the assembly of CRISPRa throughout the present study.

A previously reported tumor-specific promoter of survivin gene ($P_{Suv}$)[29,30] or a constitutive SV40 promotor was first placed upstream of the dCas9 coding cassette (Supplementary Fig. 1c, top illustration). The CRISPRa activities associated with these constructs were examined in human and mouse lung tumor cell lines (H1299 and Lewis lung carcinoma [LLC], respectively), along with the primary mouse embryonic fibroblasts (MEFs) as representatives of normal cells. The cells were transfected with plasmids encoding dCas9, MPH, and sgTGTa, together with a reporter construct with designed, 3× target sites upstream of a minimal promoter-led EGFP (see "Methods"). In reference to the activity profile of the SV40 promoter, the $P_{Suv}$ showed apparently higher activities to drive dCas9 expression in tumors cells than in MEFs (Supplementary Fig. 1c, lower, Flag panel). Importantly, the CRISPRa activities (reported by EGFP levels) in different groups defined by various cell types and dCas9 constructs closely correlated with those for dCas9 expression (Supplementary Fig. 1c, lower). These results provide evidence that controlled dCas9 expression by the use of tumor-specific promoters can enhance the tumor cell selectivity for the CRISPRa outputs.

Next, the effects of $P_{Suv}$-CRISPRa on inducing an immunoregulatory output were examined in the H1299 cell line. The endogenous IFNγ (encoded by *IFNG*), a cytokine strongly involved in natural tumor immune surveillance[31,32], was selected as an endogenous target. Guided by individual sgRNAs corresponding to different positions upstream of *IFNG*, the CRISPRa complexes (driven by $P_{Suv}$-dCas9) led to variable degrees of targeted transcription induction (Supplementary Fig. 1d). In parallel, a downstream marker of IFNγ signaling, i.e., *IRF1*, showed a similarly variable induction pattern (Supplementary Fig. 1d). Importantly, a neutralizing antibody against IFNγ largely abrogated such secondary *IRF1* induction, without affecting the CRISPRa-dependent *IFNG* activation (Fig. 1b). The introduction of CRISPRa-*IFNG* in H1299 cells also led to upregulation of cell surface HLA-ABC (Class I), which was likewise abrogated by the IFNγ-blocking antibody (Fig. 1c). These results demonstrated that an active CRISPRa SAM effector could effectively program the tumor cells for induction of endogenous IFNγ, which in turn triggered autocrine/paracrine signaling.

CRISPRa may be easily adapted for enhanced efficiency via pooled sgRNAs targeting the same gene, or for multiplexed targeting via combining more than one gene-specific sgRNAs[27,33–35]. Consistently, we found that a combination of two sgRNAs targeting *IFNG* could lead to a more robust downstream effect of HLA up-regulation (Fig. 1d). Moreover, a combination of activating sgRNAs for *IFNG* and for a T cell chemokine, i.e., *CCL21* (Supplementary Fig. 1e), could lead to simultaneous induction of both genes (Fig. 1e). Collectively, our data so far point to the potential use of CRISPRa SAM in a tumor-conditional therapeutic gene circuit, for effective and versatile induction of endogenous immune regulators.

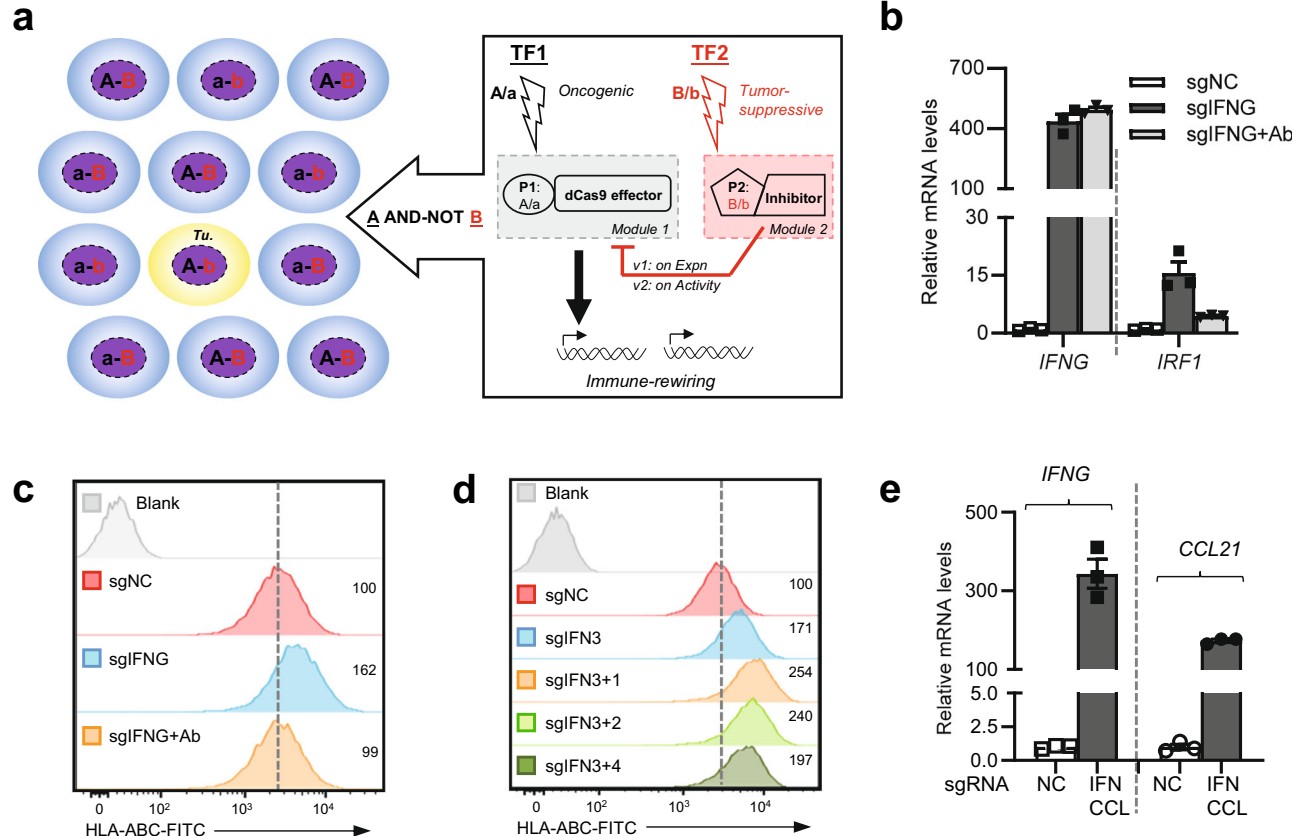

**Fig. 1 An active CRISPRa can flexibly program immunostimulatory outputs in tumor cells. a** The overall design for our tumor-targeting synthetic circuit is illustrated. Different cells on the left can be defined by the combinatorial status regarding two TFs (A/a and B/b, in black and red). The upper- and lower-case letters respectively indicate their high and low levels. A tumor cell (*Tu.*) selectively featuring simultaneous activation of TF1 and inactivation of TF2 ("A-b") is depicted in yellow. On the right, a synthetic circuit is to specify functional output in the "A-b" tumor cells. Promoters (P1 and P2) are used as sensors for TF1 and TF2, respectively. The circuit adopts CRISPR/dCas9 transcriptional effectors for input processing and actuation. Here, two sequential versions of the circuit are denoted (v1 and v2). They respectively feature a TF2-sensing inhibitory module (pink) against either the expression (v1) or activity (v2) of the dCas9 effector in the activation module (grey). **b, c** H1299 cells were transfected with dCas9 (under the survivin promoter), MPH, and a high-performance activating sgRNA against *IFNG* (sgIFNG or a negative control sgNC). Cells were incubated for 48 h in the presence of 10 μg/ml of IgG or anti-IFNγ (Ab) and were then harvested for qPCR (**b**) or flow cytometry analyses (**c**). **d** H1299 cells were transfected similarly as above for 48 h. Combinations of sgRNAs targeting sequences upstream of *IFNG* (all containing a well-performing sgRNA, i.e., sgIFN3) were included in the plasmid mix. In both (**c**) and (**d**), the dotted lines denote the levels in control cells. Additionally, the relative median fluorescence intensity [MFI] values are marked. **e** The H1299 cells were transfected similarly as in (**d**), except that sgRNAs against *IFNG* and *CCL21* were introduced for multiplexed activation. Cells were harvested 24 h after transfection for qPCR analyses. In this figure, the qPCR results are presented as mean ± SEM (*n* = 3 biological replicates). Source data are provided in the Source Data file.

**Construction of a preliminary dual-input, AND–NOT logic circuit.** Few promoters are proven to be stringently tumor-specific[36]. Therefore, the mere use of a "tumor-enhanced" promoter for dCas9 expression (as adopted in the experiments above) is most likely not sufficient to stringently ensure a tumor-specific output by the CRISPRa effector. For an improved cell specificity in output induction, we next sought to incorporate the CRISPRa effector into more sophisticated, multi-input Boolean logic circuits[4,5].

For a CRISPRa device, or likewise a CRISPRi target-repressive device[22,27,37], their individual dCas9 and sgRNA components may be placed under different TF-responsive promoters ("P1" and "P2") to enable flexible input sensing (in a singular or combinatorial format), and logic processing (Supplementary Fig. 2a). To enable Pol II-dependent sgRNA expression, a design of ribozyme-based auto-processing cassette was adopted (Fig. 2a)[38]. Using a Cas9/sgRNA-mediated cleavage reporter assay in 293T cells[39], we validated that a Pol II-type, CMV promoter-driven sgRNA had an evident targeting activity,

although at a level lower than that by conventional Pol III-type (U6) sgRNA construct. Such partially reduced activities associated with the Pol II sgRNA construct are consistent with previous reports[38,40].

Subsequently, we tested (in 293T cells) the use of Pol II promoter-driven sgRNA to shape CRISPRa/i outputs. Here, a constitutive (i.e., CMV) and a signal-dependent promoter were chosen to respectively direct the sgRNA and dCas9 moiety, or vice versa. As for the signal-dependent promoter, we selected an IFN-sensitive response element (ISRE, 5×)[41], which senses activation of an ISGF3 TF complex by type I interferon (IFN).

Specifically, in a first CRISPRi-based circuit, a CMV promoter (or a U6 promoter, as a positive control) drove the constitutive expression of a targeting sgRNA (for target inhibition, sgTGTi), while an ISRE promoter directed the expression of the KRAB repressor domain-fused dCas9 [dCas9-KRAB] (Supplementary Fig. 2b, left illustration). Alternatively, a second circuit instead incorporated an ISRE-controlled sgTGTi, together with a CMV-dCas9-KRAB (Supplementary Fig. 2c, top illustration). A repressible

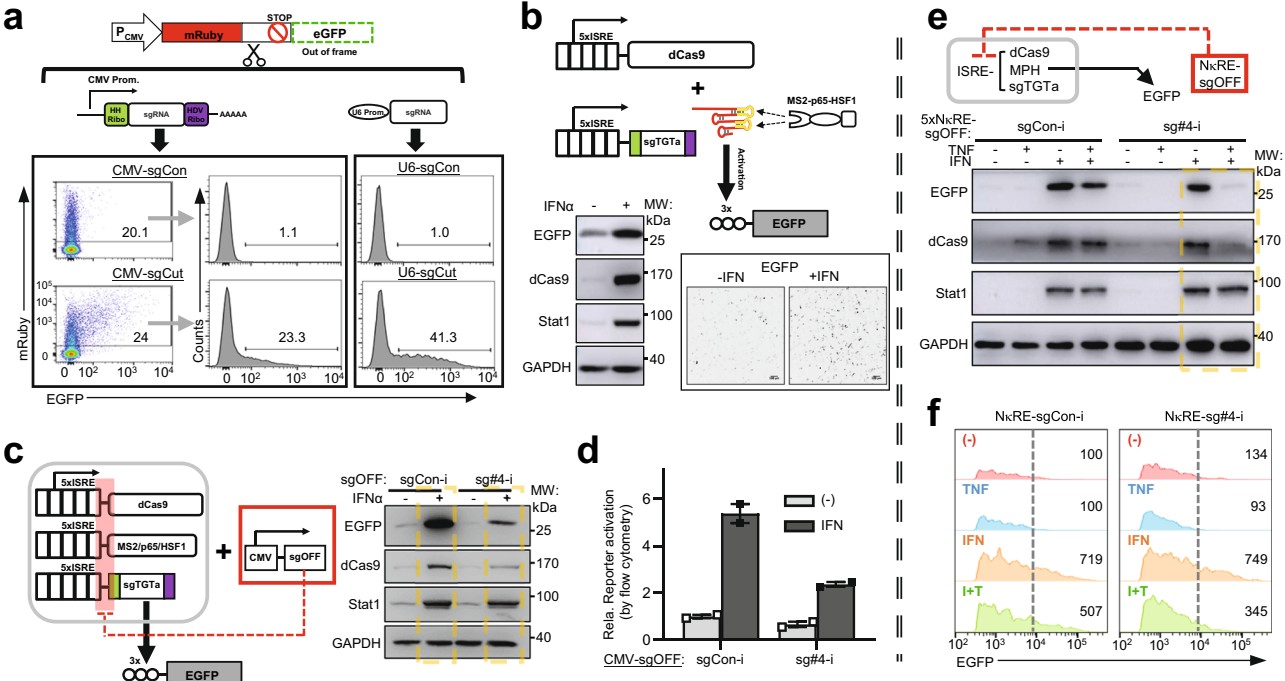

**Fig. 2 Construction of a preliminary dual-input, AND–NOT logic circuit. a** An EGFP reporter whose expression as a fusion protein would be initiated by Cas9/sgRNA-mediated upstream cleavage (frameshift) is illustrated on top. A ribozyme-based, Pol II sgRNA expression system is illustrated right below. 293T cells were transfected with Cas9 and the reporter, as well as the CMV-driven sgRNA for cutting the reporter construct (sgCUT), its Pol III counterpart, or the corresponding control non-targeting sgRNAs (sgCon). Cells were harvested (48 h) and subjected to flow cytometry. Percentages of EGFP$^+$ cells(mRuby$^+$ gate) were marked on the histograms. **b** The illustration on the top shows a Pol II promoter-controlled CRISPRa SAM complex. The dCas9 and targeting sgRNA for activation (sgTGTa) are both led by 5× ISRE, while the co-activator MPH is constitutively expressed. In the reporter, 3× target sites proceed a minimal promoter-led *EGFP*. After plasmid transfections, 293T cells were treated with human IFNα (1000 IU/ml) for 24 h. EGFP levels were determined by fluorescence microscopy (inset, scale: 100 μm), and by immunoblotting (IB, lower left). **c, d** In **c**, the illustration shows a circuit with one Pol II promoter (5× ISREs) controlling all three CRISPRa components (activating an EGFP reporter) and another (CMV) driving an off-switching sgRNA (sgOFF, targeting all three CRISPRa components). A non-targeting sgRNA (sgCon-i) serves as a control for the sgOFF. Transfected cells were treated (1000 IU/ml IFN, 48 h) and harvested for IB. In **d**, some cells were subjected to flow cytometry and followed by quantitation (EGFP$^+$%×MFI, mean ± range, n = 2 from independent experiments). **e, f** The circuit was similar to (**c**), except that the sgOFF was led by 5× NκRE. The cells were treated ±IFN (1000 IU/ml) and ±TNF (50 ng/ml) for 48 h and harvested either for IB analyses (**e**), or for flow cytometry (**f**). In **f**, the histogram shows the fluorescence pattern for EGFP$^+$ cells. The dotted lines mark high levels of EGFP positivity definitively attributed to CRISPRa activity. Relative levels of EGFP $^+$%×MFI are marked on the histogram. The IB results in **b**, **c**, **e** are representative of two independent experiments. Source data are provided in the Source Data file.

EGFP reporter (in a destabilized form, dEGFP[42]) with a unique sgTGTi-targeted sequence near the transcription start site was co-transfected for indication of CRISPRi activities. We found that both circuits were capable of programming IFN-dependent repression of dEGFP, although the outputs appeared less robust than those engaged in the U6-sgTGTi positive control group (Supplementary Fig. 2b, c). These results validated significant activities of sgTGTi driven by two Pol II promoters (CMV and ISRE), and established a platform for dynamic CRISPRi outputs via induced expression of either dCas9-KRAB or sgTGTi.

In the ensuing experiment, we constructed an IFN-signaled CRISPRa activating device (Fig. 2b, upper illustration). This device further adopted a design of co-regulation, with individual ISRE promoters driving both dCas9 and sgTGTa (besides a constitutive MPH). A CRISPRa-dependent EGFP reporter, which we used earlier (see Supplementary Fig. 1c), was monitored as circuit output. Expectedly, such a circuit programmed the cells to transduce an IFN signal into CRISPRa activation [EGFP upregulation] (Fig. 2b).

The above circuits with promoter-controlled CRISPRa/i actuators can produce outputs according to increases (gain) in TF activities. However, as TF inactivation events (loss) are prevalent in tumors, a different class of circuits whose outputs are

triggered by loss of TF (e.g., p53) activities would be particularly suitable for specific tumor targeting. This functionality requires an inhibitory module[43], whose absence allows output production (de-repression), forming a NOT-type logic control. Given the multiplexable targeting feature of dCas9, we hypothesized that the above-validated, Pol II-driven sgRNA expression system may be further harnessed for constructing a NOT logic gate. For instance, in a system with a P1 driving both a dCas9-effector and a sgTGT, the presence of an sgRNA (P2-dependent) against their promoters (with off-switching activity, sgOFF), would concomitantly engage the dCas9 moiety to form a NOT logic gate (Supplementary Fig. 2d). Note that the "sgOFF" is used as a generalized nomenclature in this work.

To test this principle, a circuit was first assembled with 5× ISRE-driven dCas9-KRAB and sgTGTi, together with a Pol III promoter-driven sgOFF (U6-sg#1-i) targeting a sequence within the ISRE promoter. Here, the sg#1-i would engage dCas9-KRAB into a dual-target repression module ("2× OFF") against itself and sgTGTi. For comparison purposes, in an alternative circuit featuring a constitutive dCas9-KRAB (CMV) and a regulated ISRE-sgTGTi, we implemented a single-target inhibitory format ("1× OFF"), where the sg#1-i was poised to suppress only the sgTGTi component (Supplementary Fig. 2e, top illustration).

With the 2× OFF format, the U6-driven sg#1-i notably impeded IFN-dependent induction of dCas9-KRAB compared to a non-targeting sgRNA (against a luciferase-derived sequence, sgLuc), which reflected the direct actions by this inhibitory module. Concomitantly, IFN-triggered synthetic output, i.e., dEGFP down-regulation, was abrogated by sg#1-i (Supplementary Fig. 2e, lower right). On the other hand, with the 1× OFF format where the U6-sg#1-i would only suppress sgTGTi, IFN-triggered dEGFP down-regulation was restored to a lesser extent (Supplementary Fig. 2e, lower left). These results suggested a framework for a sgOFF-dependent control of CRISPRi function, where a design of co-inhibition against its multiple components could lead to a more efficient control.

Unlike the U6-sg#1-i used above, a Pol II-dependent CMV-sg#1-i appeared significantly less effective in our testing attempts to suppress 5× ISRE-dependent dCas9-KRAB levels (Supplementary Fig. 2f), which was likely to reflect a combination of its dampened expression via a Pol II system and certain unfavorable features in targeting position/sequence. Subsequently, to sensitize the CRISPRi components to repression by sgOFF, we inserted artificial sgOFF target sequences (each in a 2× format) near the core promoters in both ISRE-led dCas9-KRAB and sgTGTi constructs (Supplementary Fig. 2g, top illustration). Two parallel CRISPRi actuator modules respectively featuring a different set of such artificial "cis-elements" were constructed, and were assembled together with their corresponding CMV-driven sgOFFs (sg#2-i and sg#3-i). The results showed that only one of the two CMV-driven sgOFFs (sg#3-i) visibly blunted IFN-induced dCas9-KRAB and the associated changes in dEGFP (Supplementary Fig. 2g, lower). Such variable effects by Pol II-driven sgOFFs also confirmed the importance of target optimization for their improved performance.

A CRISPRa SAM-containing circuit was also designed to incorporate a similar sgOFF-dependent inhibitory module, taking advantage of the fact that an unmodified dCas9 would serve as a common component for simultaneous gene activation (programmed with a stem–loop-modified sgTGTa) and inhibitory circuit control (with a normally scaffolded sgOFF)[37,44]. Following target optimization (Supplementary Fig. 2h), an effective CMV-driven sgOFF, i.e., sg#4-i, was assembled with the triplicate CRISPRa components which were individually led by ISRE promoters [featuring a common proximal target sequence] (Fig. 2c, left illustration). The results showed that IFN/CRISPRa-dependent EGFP reporter induction was suppressed by CMV-sg#4-i for more than 50% (compared to sgCon-i), consistent with the pattern of dCas9 expression (Fig. 2c, d).

To test the control of the sgOFF inhibitory module by a physiological signal, an NF-κB-responsive element (5× NκRE) promoter was used to drive sg#4-i. This promoter responds to NF-κB, a TF commonly activated by inflammatory signals such as TNFα. In cells co-transfected with all circuit components (ISRE-CRISPRa/sgTGTa/MPH and NκRE-sg#4-i), IFN-triggered EGFP output was evidently reduced by the co-addition of TNFα, in a pattern correlating with that of dCas9 (Fig. 2e, f). Overall, these results presented a preliminary design of a P1-CRISPRa/i effector- and P2-sgOFF-incorporated circuit to sense a TF1+/TF2− state (AND–NOT logic). In the remainder of this study, we focused on the CRISPRa SAM-featured circuits, due to their versatility and effectiveness in directly driving cells toward immune rewiring (see Fig. 1).

**Construction of a customized AND–NOT logic circuit (v1) targeting a malignant state.** The AND-NOT logic circuit is particularly relevant for driving tumor-specific output (Fig. 3a), via potentially enabling the detection of simultaneous gain- and

loss-of TF activities (e.g., hypoxia-inducible factor 1α [HIF1α] and p53, respectively) characteristic of many tumors[15–17,45]. We, therefore, sought to construct the first version of HIF1α/p53-sensing, AND–NOT logic circuit using the CRISPRa/sgOFF strategy (v1).

As the H1299 cell line is p53-deficient[46], it was used to establish the cell system for circuit testing. We validated a hypoxia-response element (HRE, 3x) in this cell line using a HIF1α-activation reagent, i.e., $CoCl_2$ (Fig. 3b)[47]. We further generated an H1299-derived stable cell line for tetracycline-controlled p53 expression and activity (p53-tet) (Supplementary Fig. 3a, b). A well-known p53-responsive promoter (from the intron of MDM2, $P_{M2}$[48]) was also validated in p53-tet H1299 cells (Fig. 3c). Individual HRE promoters were used to drive the expression of all three CRISPRa components (targeting EGFP for induction). Additionally, the core promoters for these units all contain the same sequence targeted by the sgOFF (sg#4-i), which in turn, is part of the inhibitory module controlled by a $P_{M2}$ promoter (Fig. 3d, left illustration, the circuit denoted as v1.1 to indicate its stage of development). In p53-tet H1299 cells introduced with such an AND–NOT circuit, a HIF1α input (+$CoCl_2$) engaged CRISPRa to induce EGFP expression for about fivefold under the p53-deficient condition (−DOX). In contrast, a simultaneous p53 input (+DOX) acted to partially inhibit the $CoCl_2$-signaled EGFP induction (up to ~50%), consistent with its dampening effect on dCas9-Flag protein (Fig. 3d, right).

Next, we set out to simplify the CRISPRa expression system into a single-promoter construct, as in a previous report[38]. Such simplification would also conceivably enable a more uniformed, sgOFF-mediated transcriptional suppression of all actuator components (Fig. 3e, upper illustration, circuit v1.2). Herein, a 3× HRE promoter drove a single mRNA precursor, in which dCas9 and MPH were separated by an internal ribosome entry site (IRES), followed by the auto-processing sgTGTa cassette in the 3′ untranslated region. The same $P_{M2}$-sg#4-i module was used as the NOT gate. In p53-tet H1299 cells, such a modified circuit (v1.2) wired a $CoCl_2$/HIF1α input into an about twofold increase of EGFP output in cells with the p53-null state [−DOX] (Fig. 3e). In this context, a simultaneous DOX/p53 signal restored the $CoCl_2$/CRISPRa-enhanced EGFP down to a level seen in untreated cells, suggesting an effective inhibitory control by $P_{M2}$-sg#4-i on the inducible portion of CRISPRa activity (Fig. 3e and Supplementary Fig. 3c). Nevertheless, an apparent background CRISPRa activity resisted inhibition by DOX/p53. Therefore, despite apparently exhibiting some inverse-tuning ability, the NOT logic gating of CRISPRa/i by Pol II-driven sgOFFs presented non-optimal precision in various contexts (Fig. 2d–f, Fig. 3d, e and Supplementary Fig. 2g). A further improvement in the NOT gate performance was warranted.

**Development of a more accurate NOT gate for CRISPRa by employing the anti-CRISPR AcrIIA4.** Given the observed imprecision with the sgOFF-dependent NOT gates, we realized that a pitfall of such a design is that both the actuator and its inhibitor are based on dCas9-mediated DNA targeting, which may readily endow a modest potency for the sgOFF module. The conceivably limited expression of sgOFF by a Pol II is likely to present an additional challenge for its effectiveness. Therefore, a more robust inhibitory unit against the CRISPRa would be instrumental for assembling the NOT gate. Along this line, we noted that recent pioneering works had uncovered numerous phage-derived proteins (anti-CRISPRs) that directly antagonize CRISPR–Cas activities[49]. Importantly, one of the anti-CRISPR proteins, i.e., AcrIIA4, was previously established to potently inhibit Cas9-mediated DNA sequence interrogation[24–26].

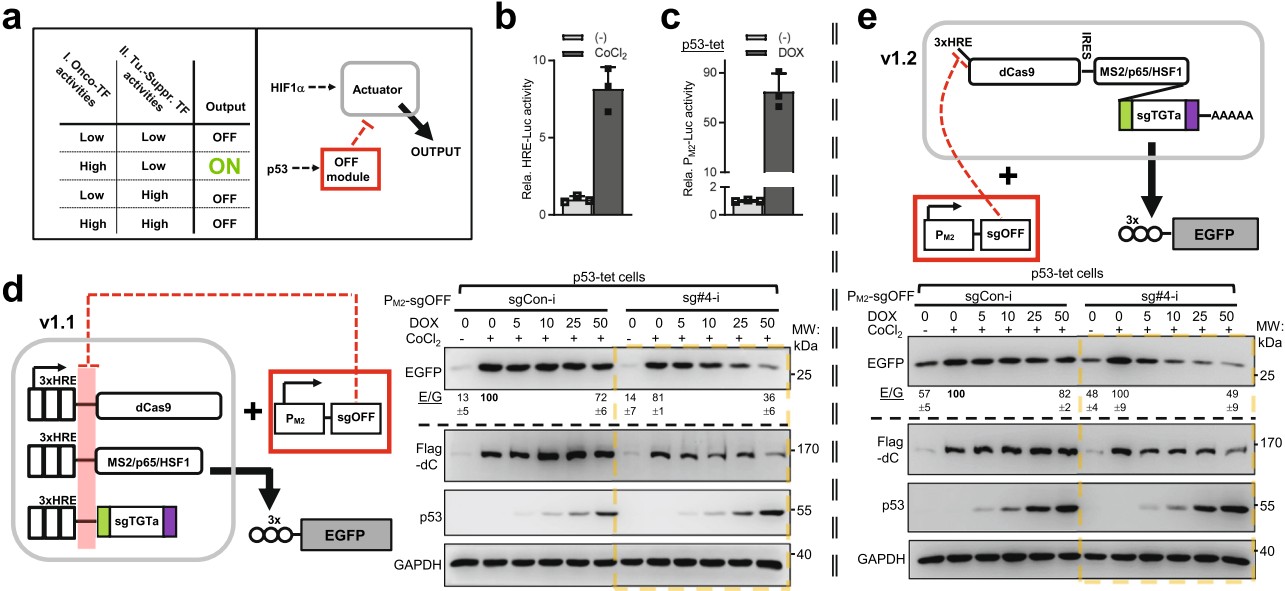

**Fig. 3 Construction of a customized AND–NOT logic circuit (v1) targeting a malignant state. a** The left box contains a two-input table based on activities of an onco-TF[(I)] and a tumor-suppressive TF[(II)]. The tumorous state can be gated by the status of this TFs using an AND–NOT logic ("ON" in green). The right box illustrates the strategy for an AND–NOT logic, tumor-targeting circuit based on the activities of HIF1α and p53. An output is engaged only in cells featuring high HIF1α activity in conjunction with p53 deficiency. **b** H1299 cells were transfected with a 3× HRE-luciferase reporter and treated with 150 μM CoCl₂ for 24 h. Relative levels of luciferase activities were presented (mean ± SD, n = 3 biological replicates). **c** H1299 cells were introduced with tetracycline-inducible p53 via a lentiviral vector ("p53-tet"). These cells were transfected with a p53-responsive $P_{M2}$-luciferase construct and treated with 50 ng/ml of DOX for 48 h. The cells were harvested for luciferase assay (mean ± SD, n = 3 biological replicates). **d** The illustration on the left shows the design for an AND–NOT circuit ("v1.1"). Each CRISPRa component (to activate EGFP transcription) is controlled individually by a 3× HRE promoter, whereas the sgOFF is driven by a p53-responsive promoter ($P_{M2}$). In the results shown on the right, the effects of $P_{M2}$-sg#4-i were compared to sg-Con-i in p53-tet H1299 cells. The circuit-introduced cells were treated with ±150 μM CoCl₂ and ±DOX (with indicated doses) for 24 h. Cell lysates were examined by IB. The dCas9 levels were represented by their Flag tag (Flag-dC). **e** All CRISPRa components were assembled into one single 3× HRE-controlled unit, as shown in the illustration on the top. Other designs and experiments were similar to those in (**d**). According to its stage of development, the circuit is named "v1.2". In **d**, **e**, the blotting results are representative of two independent experiments. In addition, quantitation for the basal, CoCl₂, and co-addition (DOX at 50 ng/ml) groups pooled from four independent experiments are marked below the corresponding EGFP panel (E/G, EGFP normalized to GAPDH, mean ± SEM). Source data are provided in the Source Data file.

Subsequently, we adopted AcrIIA4 for developing another NOT gate for CRISPRa.

We first compared the original AcrIIA4 in bacterial codons with its human codon-optimized counterpart (referred to as ACRmax herein). While both forms effectively reduced the activities of co-transfected CRISPRa, the ACRmax exhibited an even greater inhibitory potency (Supplementary Fig. 4a and Fig. 4a). Subsequently, we constructed a $P_{M2}$-ACRmax inhibitory module, which was assembled together with a single-unit $P_{Suv}$-driven CRISPRa actuator (Fig. 4b, left illustration). In p53-tet H1299 cells, the EGFP output under the p53-null condition (−DOX) was about four-fold higher than that under the DOX/p53-rescued condition (Fig. 4b, middle and right). Consistent with the mechanism of AcrIIA4 action[25], p53/ACRmax-associated regulation of EGFP output showed little correlation to the levels of dCas9. Importantly, compared to earlier experiments with the sgOFF system (see Fig. 3e and Supplementary Fig. 3c), we noticed that a lower dose of DOX (10 ng/ml) here caused a greater degree of output inhibition (Fig. 4b, right). This is not due to the use of different dCas9 constructs, as a control experiment showed that a $P_{Suv}$ or CoCl₂-activated HRE drove similar levels of dCas9 expression (Supplementary Fig. 4b). Such comparisons indicated a better performance by the $P_{M2}$-ACRmax module over its sgOFF-based counterpart. Additional tests were carried out using a p53-tet/H1299-derived sub-line with a virally integrated $P_{M2}$-ACRmax construct. Even when a strong CRISPRa (CMV-dCas9/U6-sgTGTa) actuator was introduced by transfection, engagement of the integrated $P_{M2}$-ACRmax by a DOX-restored

p53 input could effectively suppress the EGFP output (Supplementary Fig. 4c–e).

Such favorable gating performance by $P_{M2}$-ACRmax in the p53-tet H1299 cells made us proceed further. The event of p53 inactivation, although not uniformly occurring in tumors, is strongly associated with a higher degree of malignancy[50]. Therefore, it would be particularly desirable if $P_{M2}$-ACRmax inhibitory module could effectively sense the basal activity of endogenous p53, leading to accurate identification of p53-inactivated tumor cells (Fig. 4c). A549 is a human lung cancer cell line that has WT p53. The p53 functional status in these cells was additionally validated under the treatment with a p53 stabilizer (Nutlin-3) or a chemotherapeutic drug (cisplatin) (Supplementary Fig. 4f, g). We transfected both A549 and H1299 cells with a CRISPRa actuator (CMV-dCas9/U6-sgTGTa) and the $P_{M2}$-ACRmax module. In stark contrast to the unrestricted output pattern in H1299 cells, the CRISPRa-dependent EGFP output was markedly inhibited by $P_{M2}$-ACRmax in A549 cells (Fig. 4d). A similar observation was made in another p53-sufficient breast cancer cell line MCF-7 (Supplementary Fig. 4h).

To formally establish that $P_{M2}$-ACRmax-mediated inhibition on CRISPRa was responsive to basal p53 signaling in A549 cells, genetic knockout of p53 in these cells was carried out via genome editing (Cas9). Next, six different clones of parental A549 cells or the p53-deficient derivatives were introduced with the CMV-CRISPRa/$P_{M2}$-ACRmax circuit. The results indicated that targeted p53 inactivation in A549 cells markedly mitigated $P_{M2}$-ACRmax-dependent suppression of CRISPRa output (Fig. 4e).

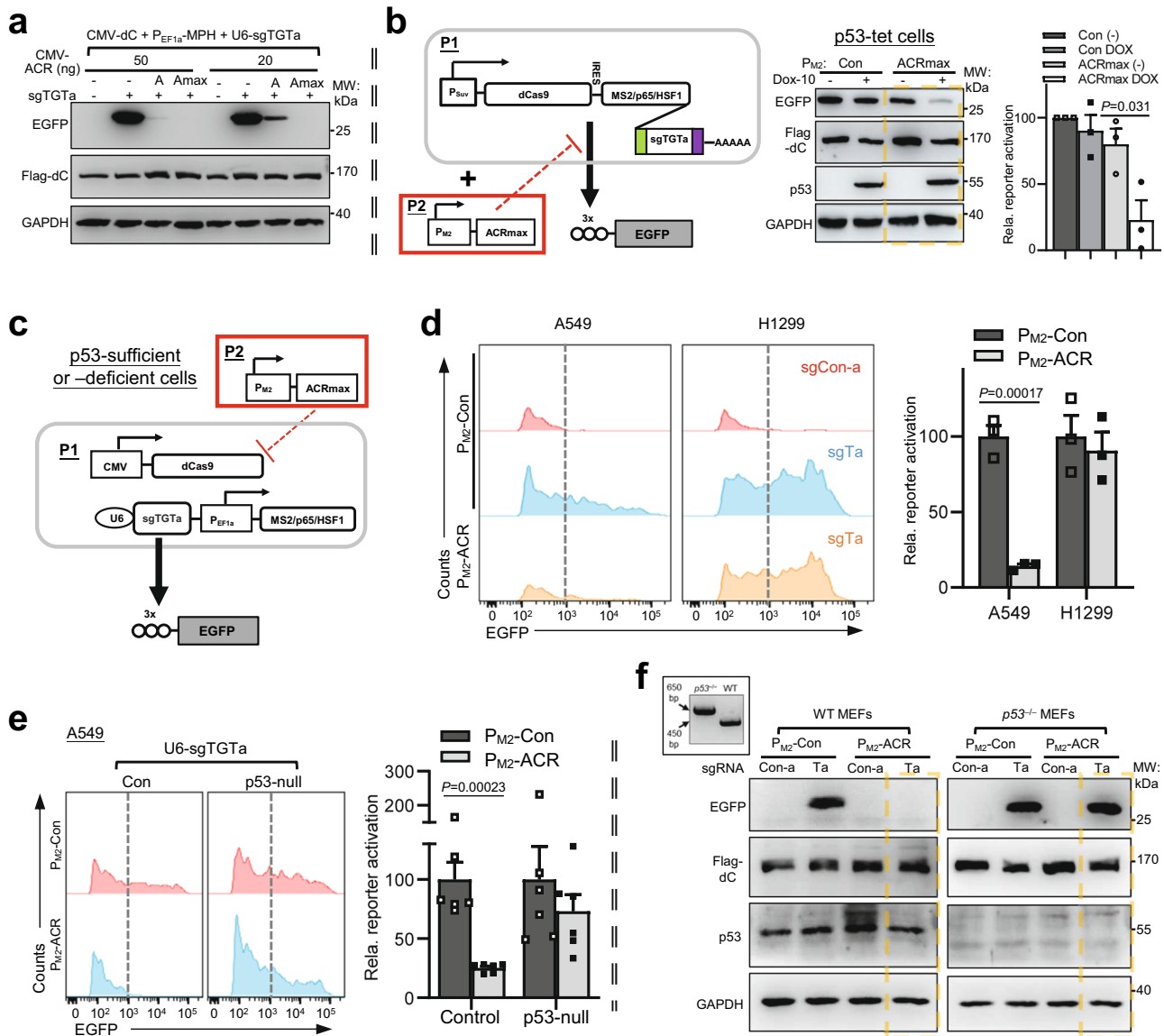

**Fig. 4 Development of a more accurate NOT gate for CRISPRa by employing the anti-CRISPR AcrIIA4. a** 293T cells were co-transfected with the AcrIIA4 in bacterial codons ["A"] or its human codon-optimized version [ACRmax, "Amax"] (both under CMV promoter), and the constitutively expressed SAM components. Thirty-six hours after transfection, cell lysates were harvested and subjected to IB for levels of EGFP and Flag-dCas9. **b** The p53-tet H1299 cells were transfected with a single promoter-driven CRISPRa expression unit targeting an EGFP reporter, and ACRmax led by $P_{M2}$ (illustration on the left, $P_{Suv}$: survivin promoter). The effects of DOX (10 ng/ml) treatment were analyzed by IB (middle). Quantitation of EGFP band intensities (normalized to those of GAPDH) is shown on the right (mean ± SEM, $n = 3$ measurements from independent experiments). **c**, **d** In **c**, the illustration shows the circuit featuring a strong CRISPRa actuator (CMV-dCas9 and U6-sgTGTa for EGFP), and an inhibitory module of $P_{M2}$-ACRmax. In **d**, the circuit was introduced into A549 and H1299 cells. A non-targeting sgRNA (sgCon-a) was the control. A representative histogram shows the fluorescence pattern for EGFP$^+$ cells ("Ta": sgTGTa). The dotted lines mark high levels of EGFP positivity definitively attributed to CRISPRa. The quantitation is shown next to the histogram (EGFP$^+$%×MFI, mean ± SEM, $n = 3$ biological replicates). **e** Isogenic clones (six each) of WT or p53-deficient A549 cells were prepared after CRISPR/Cas9-mediated genome editing. The circuit in **c** was introduced into the cells and their EGFP signals were determined by flow cytometry (left). The dotted lines highlight p53/$P_{M2}$-ACR-driven inhibition of CRISPRa activities. The quantitation for fluorescence (EGFP$^+$%×MFI) is presented on the right (mean ± SEM, $n = 6$ independent WT and p53$^{-/-}$ cell lines transfected in parallel). **f** WT or p53$^{-/-}$ MEFs were prepared. The inset shows a representative genotype analysis. The circuit in **c** was introduced to the cells (sgCon-a as a control). The cell lysates were subjected to IB. One-sided Student's $t$-tests were used for statistical analyses in this figure ($P$ values provided). Blotting results in (**a**–**c**) are representative of 2, 3, and 2 independent experiments, respectively. Source data are provided in the Source Data file.

We also conducted similar tests in primary MEFs, taking advantage of the *p53* germline knockout mouse model. A clear de-repression of EGFP output was observed in *p53*$^{-/-}$ cells, in comparison to the WT MEFs (Fig. 4f). Interestingly, further experiments showed that a $P_{M2}$-ACRmax inhibitory module was engaged even in *p53*$^{+/-}$ MEFs, leading to marked suppression of

the EGFP output driven by a constitutive SV-40-dCas9 (Supplementary Fig. 4i). These results from the isogenic control and p53-null cells demonstrate that the $P_{M2}$-ACRmax module was operated by basal p53 signal (even with a single allele), and consequently enabled stringent gating of CRISPRa output only by severe p53 deficiency. It is also encouraging that in these

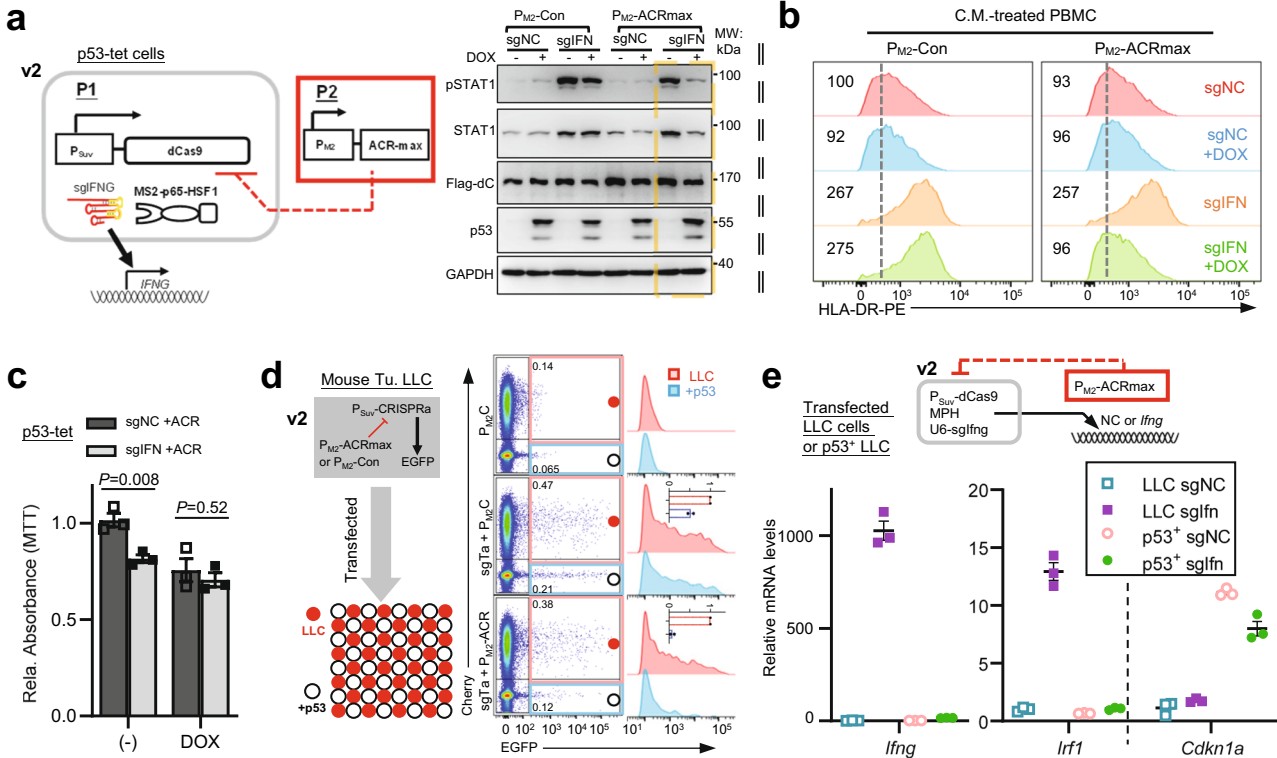

**Fig. 5 An improved $P_{Suv}/P_{M2}$ AND-NOT logic circuit (v2) rewires p53-deficient tumor cells to produce IFNγ. a–c** In **a**, the p53-tet H1299 cells (±10 ng/ml DOX) were introduced with the circuit shown on the left ("v2"). The $P_{Suv}$-driven CRISPRa is programmed to activate transcription of endogenous *IFNG*, whereas $P_{M2}$-ACRmax forms an inhibitory module (compared to a non-expression construct, $P_{M2}$-Con). The non-targeting sgNC was used as a negative control for circuit actuation. The cell lysates were harvested 48 h after transfection and were subjected to IB. The data shown are representative of two independent experiments. **b** The conditioned media from the cells were also collected and were added to freshly prepared PBMC for 24 h. The cells were subjected to flow cytometry of Class II HLA levels (CD45$^+$CD11b$^+$-gated). The relative MFI values are marked on the histograms. The dotted lines denote the control levels. In **c**, p53-tet cells were transfected with the circuit at low confluency (~20%). For target activation, a construct containing a tandem of U6-dependent IFNG-targeting sgRNAs (sgIFN3 + 1) was used. Ninety-six hours after transfection, the cells were subjected to MTT assay. A quantitative summary from three independent experiments is presented (mean ± SEM, two-sided *t*-test, *P* values provided). **d** Co-cultures containing LLC cells (p53-deficient, labeled with mCherry) and their p53$^+$ knock-in derivatives in equal proportion were established. The mixed cells were transfected with the $P_{Suv}$-CRISPRa/$P_{M2}$-ACRmax circuit (v2) targeting EGFP (sgTa: sgTGTa) for 24 h. Cells were subjected to flow cytometry analyses for mCherry and EGFP fluorescence (pseudo-color). The EGFP$^+$ subpopulation in either the mCherry$^+$ and mCherry$^-$ gates are further resolved in an EGFP fluorescence histogram, respectively in red and blue. The insets in the histograms show the averaged EGFP levels (EGFP$^+$%×MFI) in these different gates from two independent experiments. **e** The parental and p53$^+$ LLC cells were respectively transfected with the $P_{Suv}/P_{M2}$ circuit (v2) for conditional mouse *Ifng* activation by CRISPRa. The mRNA levels for IFNγ and its target gene *Irf1* are shown. *Cdkn1a* levels report p53 status. These qPCR results are presented as mean ± SEM (n = 3 biological replicates). Source data are provided in the Source Data file.

experiments, the basally engaged $P_{M2}$-ACRmax module was sufficiently potent in relation to a strong CRISPRa output (driven by constitutive dCas9/U6-sgTGTa), which in turn would be favorable for driving therapeutic activities.

**An improved $P_{Suv}/P_{M2}$ AND–NOT logic circuit (v2) rewires p53-deficient tumor cells to produce immunostimulatory ligands.** Given the establishment of a stringent p53-responsive NOT gate for CRISPRa activity, we proceeded to establish an improved AND–NOT tumor-targeting circuit (v2) and to additionally wire it toward an immunotherapeutic output (endogenous IFNγ). To this end, a tumor-preferential CRISPRa module ($P_{Suv}$-dCas9/U6-sgIFNG) was assembled with a $P_{M2}$-ACRmax inhibitory module (Fig. 5a, left illustration). Following transfection to the p53-tet cells, such a circuit (v2) enabled a marked activation of the IFNγ-STAT1 signaling in p53-null, but not p53-rescued (+DOX) cells (Fig. 5a, right). Additionally, the conditioned medium from p53-null, but not p53-rescued cells showed apparent paracrine activities to upregulate HLA proteins in tumor

cells (Class I HLA, in H1299 cells), and in the freshly prepared peripheral blood mononuclear cells (Class II HLA) (Supplementary Fig. 5a and Fig. 5b). Furthermore, consistent with the reported direct actions of IFNγ on tumor cell growth[51], noticeable circuit-engaged growth-inhibitory effects (triggered by two *IFNG*-activating sgRNAs in combination) occurred selectively in p53-null (−DOX) cells (Fig. 5c). The circuit could also be conveniently adapted to target p53-deficient cells for multiplexed induction of immune stimulators, i.e., IFNγ and CCL21, via co-delivery of their corresponding activating sgRNAs (Supplementary Fig. 5b).

The mouse LLC cell line is p53-deficient[52,53] and represents another suitable model to examine the p53 loss-gated tumor rewiring by our circuit. For comparison purposes, we established an LLC-derived knock-in line (p53$^+$), where the WT *p53* coding region was re-introduced into the endogenous locus to rescue the bi-allelic point mutations (a nonsense mutation at E32 and an R334P mutation, respectively) in the parental LLC cells (Supplementary Fig. 5c, d). The resultant p53$^+$ line showed moderately increased levels of basal and Nutlin-enhanced p53 activities

measured via a transient $P_{M2}$-EGFP reporter (Supplementary Fig. 5e).

The parental LLC and their p53$^+$ derivatives were next used to further test the cell-selectivity for the synthetic output by a $P_{Suv}$/$P_{M2}$ AND-NOT circuit (v2). Here, the choice of $P_{Suv}$ for driving CRISPRa was based on our earlier comparative experiments in the LLC tumor cells and MEFs (Supplementary Fig. 1c). In an application context, it is desirable that an evenly distributed gene circuit can restrictively trigger outputs in the targeted subpopulation. In a proof-of-concept experiment, we established a co-culture of LLC cells and their p53$^+$ derivatives, respectively representing the targeted and non-targeted subpopulations. For easy distinguishment of the two subpopulations, the parental LLC cells had been labeled in advance with mCherry. Subsequently, the co-culture was transfected with the $P_{Suv}$/$P_{M2}$ AND–NOT circuits (v2) and an output reporter [EGFP] (Fig. 5d, left illustration). Indeed, with the mock NOT gate, the $P_{Suv}$-CRISPRa-dependent EGFP output showed a similar order of magnitude in the two subpopulations (Fig. 5d, right panel). In contrast, the inclusion of the $P_{M2}$-ACRmax NOT gate largely prevented CRISPRa activation in the p53$^+$ LLC derivatives (mCherry$^-$), which led to the specification of the p53-deficient LLC subpopulation (mCherry$^+$) for an EGFP output (Fig. 5d). These results indicate the capability of a transfected $P_{Suv}$-CRISPRa/$P_{M2}$-ACRmax circuit for stringent identification of the p53-deficient subset within a cell population.

Analogous to earlier experiments in the human-derived p53-tet cells (Fig. 5a), the selectivity by a $P_{Suv}$/$P_{M2}$ AND–NOT IFNγ-inducing circuit (v2) for p53-deficient cells were additionally tested in the mouse LLC cells and their p53$^+$ derivatives (Fig. 5e). To this end, activating sgRNAs for mouse *Ifng* (two pre-screened sgRNAs cloned in tandem) was adopted. As expected, such a circuit selectively enabled the p53-deficient, parental LLC cells to robustly induce their expression of IFNγ and its target gene *Irf1* (Fig. 5e).

**The $P_{Suv}$/$P_{M2}$ AND–NOT circuit (v2) empowers immune rewiring of p53-deficient tumors, driving inhibition of tumor progression in vivo.** To determine the specificity and activity of the above logic-gated *Ifng*-inducing gene circuit (v2) at a global level, we performed RNAseq analyses on circuit-transfected LLC cells and their p53$^+$ derivatives. The two cell lines showed differences in basal levels of hundreds of genes, most of which possessed a grouped pattern of higher expression in the p53$^+$ cells (Supplementary Fig. 6a, top red box). A number of classical p53 targets were identified in this group (Supplementary Fig. 6b)[54], consistent with the reconstitution of p53 activity in the knocked-in cells. Notably, the logic circuit strongly induced a different gene set in the parental, but not the p53$^+$ cells (Supplementary Fig. 6a, green box). The principal component analyses also illustrated such a parental cell-selective regulatory program by the logic circuit (Fig. 6a). A closer examination confirmed the induction of *Ifng*. Importantly, the vast majority of the circuit-induced genes are known downstream targets of IFNγ (Fig. 6b, annotations in orange)[55], confirming the specificity of a CRISPRa-driven transcriptional rewiring. Overall, this gene set shows strong enrichment for biological functions such as cytokine actions, immune responses, and antigen presentation/processing (Supplementary Fig. 6c). These unbiased analyses demonstrate that our AND–NOT logic circuit (v2) can empower a highly specific and effective tumor recognition/immune rewiring axis with therapeutic implications.

Effective therapeutic tumor targeting by a gene circuit is likely to require a viral vector-mediated delivery method. As a prototype for viral circuit delivery, we constructed a mix of two lentiviral vectors

(non-fluorescent, LV$_{NF}$) incorporating the $P_{Suv}$/$P_{M2}$-directed IFNγ-inducing circuit [v2] (Supplementary Fig. 6d, left illustration). As expected, in a co-culture system with p53-deficient and -sufficient LLC cells (see Fig. 5d), the virally delivered circuit also exhibited an evident cell-state selectivity and drove a much greater induction of IFNγ in the p53-deficient subpopulation [mCherry$^+$] than in its p53-sufficient counterpart (Supplementary Fig. 6d, right panel).

Tumor cells introduced with the gene circuit through lentiviral vectors would enable investigations regarding the therapeutic potential of the circuit in an in vivo context. For this line of investigation, additional development was made to adapt the IFNγ-inducing circuit v2 into another mix of two lentiviral vectors with fluorescent labels (LV$_{FL}$), which would allow the assessment of transduction efficiency. As the circuit was divided into two units of LV$_{FL}$, full circuit delivery to the cells would be indicated by co-labeling of markers [mCherry and EGFP] (Supplementary Fig. 6e, left illustration). A circuit with a non-targeting sgRNA was used as the negative control. With a single round of transduction in culture, about 6% of LLC cells, and to the same extent, the p53$^+$ cells were introduced with the complete functional circuit (Supplementary Fig. 6e, the mCherry$^+$EGFP$^+$ cells in the sgIfn groups). As expected, the circuit-transduced parental LLC cells displayed a robust IFNγ signal (Supplementary Fig. 6f). On the other hand, no changes in the in vitro growth characteristics were observed for these particular cells (Supplementary Fig. 6g). Subsequently, the circuit-transduced LLC and p53$^+$ cells were transplanted subcutaneously into syngeneic mice (C57/BL6) for assessment of tumor progression in vivo. Importantly, the circuit-introduced parental LLC tumors, but not their p53$^+$ counterparts, showed clear inhibition in progression (Fig. 6c). A few (two out of ten) tumors in this group even failed to become palpable. Since the circuit-induced IFNγ did not affect the growth of these cells in vitro (see above), such effects in vivo is conceivably attributed to stimulation of anti-tumor immunity. Consistent with this notion, in tumor samples harvested at the endpoint (day-17), the circuit-introduced parental LLC group showed substantial increases in not only the direct immunostimulatory targets of the IFNγ axis (e.g., STAT1 and MHC molecules), but also markers for T cell activation and their cytotoxic function, including the mRNAs for CD25, CD69, Granzyme B and Perforin (Fig. 6d). Interestingly, the levels of *Cd8a* mRNA, indicative of the abundance of CD8$^+$ T cells did not show similar changes in these samples. Flow cytometry analyses of tumor tissues at day-7 and day-17 also failed to reveal significant changes in the numbers of CD8$^+$ T cells in response to the circuit activities (Supplementary Fig. 6h). Therefore, the associated enhancement of cytotoxic T cell activities, rather than their abundance, is most likely to underlie the therapeutic effect by our p53 loss-gated, IFNγ-inducing circuit. Collectively, these results from different levels suggest the potential of our AND–NOT logic-controlled CRISPRa circuit as a useful tool for tumor-specific delivery of immunological interventions.

## Discussion

Current cancer immunotherapy regimens and drug candidates are mostly applied via systemic administration. Such conventional treatments are often associated with toxicities, which may limit drug dosages to cause sub-optimal efficacies[3]. Therefore, the development of strategies for tumor-specific delivery of immunotherapeutic activities shall contribute to new treatment opportunities for cancer patients. Recently, the CRISPRa platform has emerged as a powerful tool to program immune activation[20,23,56,57], which underscores its potential in immunotherapeutic applications. Here, we sought to assemble a class of CRISPRa-based immunostimulatory circuits,

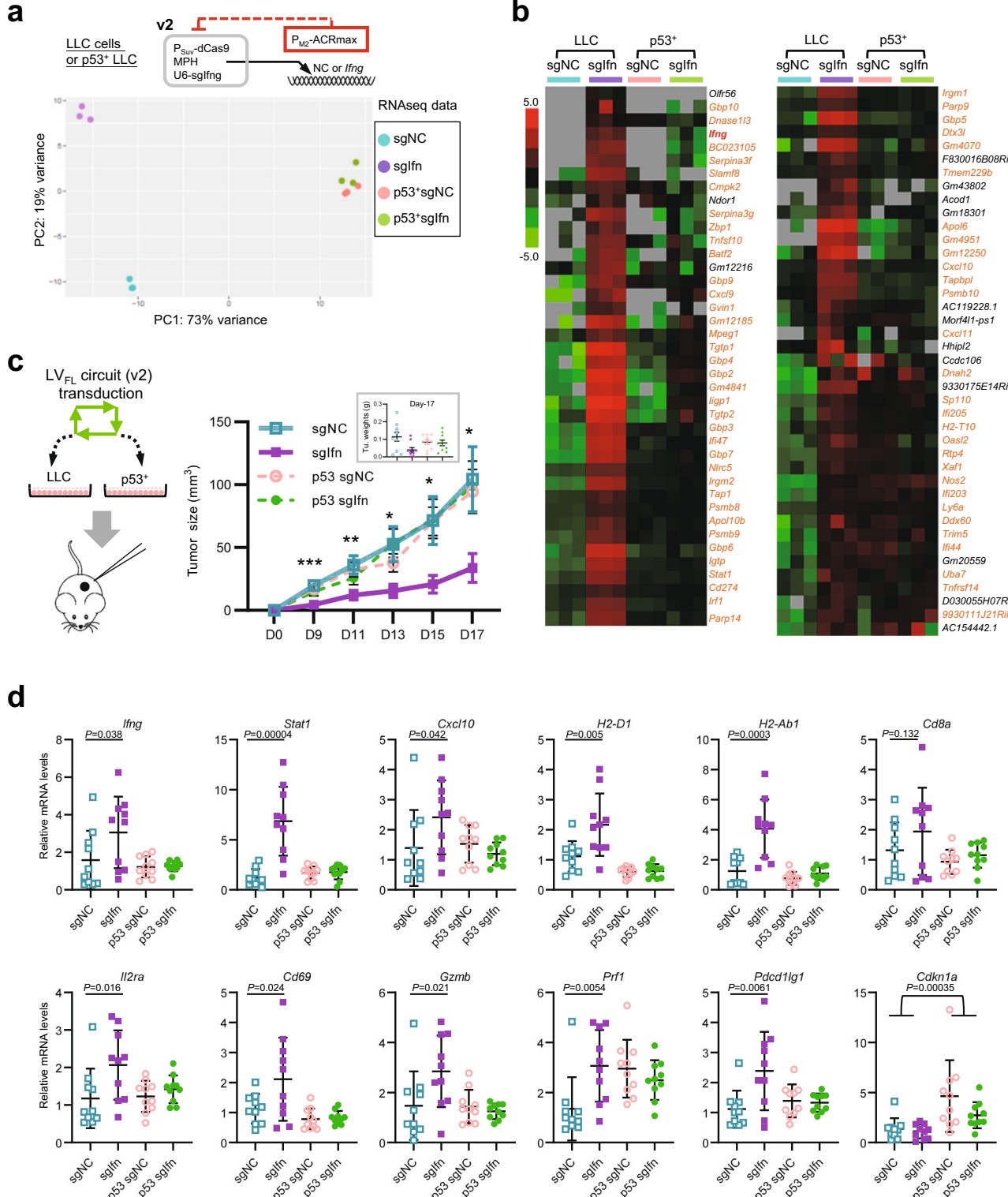

which could enable accurate tumor targeting upon co-detection of the highly tumor-characteristic p53 deficiency together with a frequent event(s) of TF activation (in a dual-input, AND–NOT logic). Two TF-regulated (via promoters) modules, respectively employing a CRISPRa actuator and a cognate off-switch, were incorporated in the circuit to identify such simultaneous gain and loss of TF activities in tumors (Fig. 1a and see Supplementary Fig. 1a for the overall flow of experiments).

We first validated that a CRISPRa/i device may be controlled by the promoter-gated expression of the dCas9 part and/or the sgRNA (Supplementary Fig. 1c and Fig. 2). Such a control approach (e.g., via $P_{Suv}$-directed CRISPRa) may be readily used for sensing heightened onco-TF activities in tumors. Intriguingly, careful examination of CRISPRa/i devices driven by a classic signal-stimulated promoter, i.e., 5× ISRE, has revealed some background activities reflective of basally expressed dCas9 part or sgRNA (Supplementary Fig. 2b, c), which was still observed for a

**Fig. 6 The $P_{Suv}/P_{M2}$ AND–NOT circuit (v2) empowers immune rewiring of p53-deficient tumors, driving inhibition of tumor progression in vivo.**
**a**, **b** The parental and p53$^+$ LLC cells were respectively transfected with the $P_{Suv}/P_{M2}$ circuit (v2) for conditional mouse *Ifng* activation by CRISPRa. The RNA samples were subjected to RNAseq analyses. **a** The principal component analyses were performed on the datasets. **b** The genes with circuit-dependent induction (FC ≥ 4, $P_{adj}$ < 0.05) in parental LLC cells are selected. Their overall expression patterns are shown in a heatmap. The annotation for *Ifng* is highlighted in red, whereas the known IFNγ targets are highlighted in orange. **c** As illustrated on the left, the $P_{Suv}$-CRISPRa-*Ifng*/$P_{M2}$-ACRmax circuit (a circuit with sgNC as a control) was packaged using a mix of two lentiviral vectors with fluorescent labels ($LV_{FL}$), where the fluorescent labels would enable determination of transduction efficiency. The LLC cells and their p53$^+$ derivatives were respectively transduced in vitro. After determination of transduction efficiency (see Supplementary Fig. 6e), the unselected cells were implanted subcutaneously to the flanks of mice (2 × 10$^6$). The trends of tumor growth [size] are shown on the right ($n$ = 10, mean ± SEM, two-sided *t*-tests performed between the sgNC and sgIfn groups at different time points). Asterisks are used to denote statistical significance (*$P$ < 0.05; **$P$ < 0.01; ***$P$ < 0.001), and the exact $P$ values between the two groups on day-9, 11, 13, 15, 17 are 0.00053, 0.0095, 0.022, 0.022, and 0.022, respectively. The inset shows the individual tumor weights determined at the harvest. **d** The above tumor samples were subjected to preparation of total RNA. The RNA samples corresponding to individual tumors were analyzed for the levels of indicated markers via qPCR analyses ($n$ = 10, mean ± SEM). One-sided Student *t*-tests were performed. Some apparent differences in markers are found between the two groups of LLC tumors (except for *Cd8a*), while parallel comparisons between the two groups of p53$^+$ tumors do not show significant differences. Source data are provided in the Source Data file.

CRISPRa effector even with dCas9 and sgTGTa each under a 5× ISRE (Fig. 2b). Placement of the MPH component also downstream of a copy of inducible promoter appeared to reduce such basal activity (Fig. 2c, e and Fig. 3d). Nevertheless, in a more applicable, single-promoter/transcript format, background CRISPRa activation was indeed apparent, pointing to its persistent nature (Fig. 3e). Such basal CRISPRa activities pose additional safety concerns, besides those apparently associated with the imprecision of "tumor-specific" promoters[12]. This further highlights the advantages in our overall design (AND–NOT logic) to improve the gene circuits' targeting specificity, as the inhibitory module in the confirmatory NOT gate would act to suppress the basal, as well as promoter non-specificity-related CRISPRa activation in non-targeted cells.

Toward a major aim of developing NOT gates for CRISPR effectors, we initially adopted an inhibitory sgRNA (sgOFF)-based strategy. This design exploited the multiplexable targeting ability of dCas9 to establish negative regulation against the levels of actuator components. Here, besides the sgOFF, the sgTGTa/i was also introduced in a Pol II-format (instead of a more efficient Pol III construct), in an attempt to enhance the adaptability of the actuator to a NOT gate control (Fig. 2). With this design principle, a class of HIF1α/p53-regulated tumor-targeting gene circuits (AND-NOT logic, denoted as "v1") was also constructed (Fig. 3). In a number of trials, we demonstrate that the expressed sgOFF may reduce CRISPRa/i activities. However, some residual CRISPRa/i activities were resistant to the blockade by sgOFF, despite significant optimization efforts (Figs. 2 and 3 and Supplementary Figs. 2 and 3). The non-optimal inhibitory potency by sgOFF can be attributed to the fact that both the actuator and the off-switch involve the same operative mechanism. Under the current Pol II-driven, less robust sgOFF expression condition, such a modest inhibitory switch would be insufficient to form a stringent NOT gate. Nevertheless, as an sgOFF is an RNA-only unit, and acts to limit the expression of dCas9 (i.e., a foreign protein) in non-targeted cells, a CRISPR effector circuit gated by such a unit may potentially possess certain advantages such as lower immunogenicity. Therefore, future efforts to improve this type of NOT gates are still warranted. This may entail innovations to enhance Pol II-mediated sgOFF expression, as the effects by a pilot circuit adopting the conventional U6-driven sgOFF have suggested (Supplementary Fig. 2e).

Our further quest for a better-performing NOT gate led to a second strategy based on codon-optimized AcrIIA4 (ACRmax), a highly potent, direct CRISPR/Cas9 inhibitor. Notably, a circuit consisting of CRISPRa and such an improved p53-sensing NOT gate ($P_{M2}$-ACRmax) enabled selective output production in the p53-deficient tumor cells even under basal culture conditions

(Fig. 4). Furthermore, such a stringent NOT-gating was achieved under the context of a strong actuator (e.g., CMV-dCas9/U6-sgTGTa), which indicates the circuit's good dynamic range suited for elicitation of a robust functional output in targeted cells. Although a circuit adopting a stringent p53 NOT gate would have the limitation to require severe p53 deficiency for targeting (thus unable to recognize some tumors retaining p53 or featuring its partial loss-of-function), it would present a lower risk for undesired activation in normal cells, which is a key goal for the current development of therapeutic circuits[4–6]. For a safe application in vivo, it would be rational to next determine whether a $P_{M2}$-ACRmax module is sufficiently sensitive to distinguish the p53-null from a physiological p53-low state (e.g., in highly differentiated cells) within a tissue context[58–60]. In the event that further improvement of the NOT gate would be required, a potential strategy to supplement a $P_{M2}$-ACRmax inhibitory module with a sgOFF unit could be adopted. Additionally, it is also pertinent to reiterate that an onco-TF-controlled CRISPRa (e.g., $P_{Suv}$-CRISPRa adopted herein, see validations in Supplementary Fig. 1c) can provide an independent level of tumor selectivity in addition to that shaped by a $P_{M2}$-ACRmax NOT gate (in an AND–NOT logic), to reduce the risk of possibly targeting normal cells.

For effective activation of anti-tumor immunity, the present work focused on the well-established IFNγ pathway, whose activation in both tumor and immune cells in the microenvironment constitutes a key part of the natural tumor immune surveillance program[31,32,51]. Therefore, we further programmed our improved AND–NOT tumor-targeting circuit toward induction of IFNγ, whose intratumoral production/function may mimic such a protective axis (Figs. 5 and 6). Notably, the transcriptome analyses not only confirmed the circuit's output selectivity for p53-deficient cells but also established the strong target (IFNγ)-specificity associated with the output (Fig. 6a, b). Such exceptional specificity on gene regulation is in line with previous reports on CRISPRa/i[27,61]. As an additional note, it may be speculated that our CRISPRa-incorporated circuit could potentially hold advantages in avoiding excessive immune activation, as previous investigations have shown that CRISPRa-mediated endogenous gene induction is more physiological (of lower magnitude) than cDNA-mediated overexpression[27].

For a model of therapeutic tumor targeting, we resorted to mice implantation experiments using circuit-introduced tumor cells. Particularly, as a prototype for viral circuit delivery, the lentiviral vectors were adopted for the introduction of a $P_{Suv}$-CRISPRa-*Ifng*/$P_{M2}$-ACRmax circuit to LLC tumor cells and their p53 knocked-in derivatives. It is worth noting that no preselection of viral transductants was conducted prior to tumor

implantation, which is relevant to an application context. Even under partial delivery, the circuit-introduced parental LLC cells, but not their p53$^+$ derivatives, showed substantially inhibited progression in vivo, which is associated with an evident engagement of anti-tumor immunity (Fig. 6d). Interestingly, the targeted LLC tumors also showed an increase in *Pdcd1lg1* mRNA encoding PD-L1 (Fig. 6d), which is consistent with a paradoxical role of IFNγ in inducing certain immunosuppressive signals[62,63]. The latter observation suggests a logical strategy to potentially combine the application of our tumor-specific immune-rewiring circuit with the existing checkpoint inhibition modalities.

Besides rewiring the tumors toward producing soluble immune stimulants, the further adaption of our circuit to activate other types of endogenous targets (e.g., cell surface stimulatory molecules and immunoregulatory TFs) would also provide exciting opportunities for advancing both basic and applied tumor immunology. Nevertheless, from an application perspective, we recognize that safe and effective means to deliver the present CRISPRa-based tumor-rewiring circuits in vivo remains a significant challenge, especially given their large sizes[5]. Future explorations through the use of the clinically relevant viral vectors such as adeno-associated virus vectors, oncolytic viruses, and other cutting-edge delivering technologies are highly warranted to study the translational potential of the presently developed gene circuits[23,56,64,65].

Although not all tumor cells would universally feature deficiencies in the p53 pathway (e.g., A549 and MCF-7 cells), tumors with p53-null (or severely deficient) status are strongly associated with high malignancy, poor prognosis, and treatment resistance[50]. On the other hand, p53-deficiency causes genome instability and is conducive to the accumulation of neoantigens[66]. Therefore, selective immune rewiring of p53-deficient tumor cells may potentially exploit some hidden vulnerability for this large group of refractory tumors to reinvigorate an antigen-dependent adaptive immunity for overall treatment efficacy. By the same token, for heterogeneous lesions with potentially mixed p53 statuses, an initial targeting of p53-deficient tumor cells may trigger an enhanced immune attack against antigens shared by all tumors. Taken together, the tumor-targeted immune-rewiring genetic circuit presented here offers a potent, versatile and safe cancer therapeutic platform that warrants active explorations.

## Methods

**Ethical statements**. This study complies with all relevant ethical regulations. The animal care and use protocols were in strict accordance to the Regulation for Management of Laboratory Animals (1988) and Guidelines for Care and Use of Laboratory Animals (2006) in China. The animal experiments were approved by the Institutional Animal Care and Use Committee of the Model Animal Research Center of Nanjing University (MARC-NJU).

**Plasmids and reagents**. The non-plasmid reagents (e.g., antibodies, cytokines, and cell lines) are listed in Supplementary Table 1. The plasmids were constructed using conventional restriction enzyme cloning (Takara, T4 DNA Ligase, #2011A) or recombinase-based cloning (Vazyme, ClonExpress II One Step Cloning Kit, #C112-02-AB). The key plasmid sequences are shown in Supplementary Data 1. The sequences for the ISRE, NκRE, HRE, the tumor-specific survivin promoter (−205 to +64), and the p53-responsive *MDM2* promoter (within intron 1) were described previously[29,30,41,48,67,68]. A CMV expression plasmid (pST1374) was used as a backbone for dCas9-KRAB or dCas9 constructs, with different promoter sequences (5× ISRE-miniCMV, 3× HRE-miniCMV or P$_{Suv}$) were cloned in place of the original CMV promoter. The pLKO.1 construct (U6-sgCUT) and pGL3-backboned (addgene: #51133) constructs (sgLuc and sg#1-i) were used for Pol III-dependent sgRNA expression. For activation of endogenous *IFNG* (or *Ifng*), constructs with sgRNA cassettes placed in tandem were used in some experiments. The inhibitable dEGFP reporter was constructed using a pCMV backbone, where the target site for sgTGTi (orthogonal in the same circuit) covers part of mini-CMV and part of its featured downstream cloning sequence. The activatable EGFP reporter was based on a pGL3 backbone, a repeat of three target sites for sgTGTa (orthogonal in the same circuit) was cloned upstream of the miniCMV sequence

preceding EGFP (featuring minimal background signal, as shown previously with a similar construct[69]).

The Pol II promoter-driven sgRNA expression systems were constructed using a pGL3 backbone. Different sgRNA expression units with a sgRNA scaffold flanked by HH and HDV ribozyme sequences were cloned downstream of a CMV promoter (sgCUT, sgCon-i, sgTGTi, and sgOFFs including sg#1-i to #5-i). The CMV-sgOFFs remain orthogonal to their designed targets, which are either artificially inserted sites (#2, #3), or sequences unique to given constructs (#4 and #5 in the 5× ISRE-constructs, the former also present in the 3× HRE-constructs). The 20-bp targeting sequence for sg#4-i covers part of the miniCMV sequence as well as downstream featured cloning sequence in 5× ISRE- and 3× HRE-dCas9. In some experiments, 5× NκRE-miniCMV or P$_{Mdm2}$ (P$_{M2}$) was used to replace CMV to drive sg#4-i expression.

For the expression of gene-activating sgRNAs, a pGL3-based activating U6-sgRNA construct already containing a P$_{EF1a}$-MPH unit was used as a backbone. For Pol II-mediated expression, the HH and HDV ribozyme sequences on each side of an MS2-recruiting sgRNA scaffold (sgTGTa) were first cloned. Promoter sequences identical to those in the dCas9 constructs (i.e., 5× ISRE- and 3× HRE-dCas9, similarly targeted by sgOFF) were then cloned upstream of the sgTGTa. In some experiments with regulated MPH expression, the same 5× ISRE-miniCMV sequence as above was used to replace the EF1a promoter.

The *AcrIIA4* was synthesized according to the sequence from a previous report[26]. An online program by GenScript was used (https://www.genscript.com.cn/gensmart-free-gene-codon-optimization.html) for codon optimization. The moderately and highly optimized cDNA sequences were then added with a C-terminal nuclear localization sequence (SV40 NLS) to establish the AcrIIA4 and ACRmax cassettes, respectively. These were then cloned downstream of CMV or P$_{M2}$ within a pGL3 backbone.

The human *TP53* cDNA was cloned using pTRIPZ vector, in place of the original RFP-shRNAmir cassette. Therefore, the cDNA is driven by 7× TetO, and its expression is subjected to induction by doxycycline treatment. In another lentiviral construct, the p53-responsive, P$_{M2}$-ACRmax cassette was cloned in lentiCRISPR V2 (addgene, #52961) in place of the CMV-Cas9 cassette.

**Designing targeting sgRNAs**. The Cas-OFFinder online tool (http://www.rgenome.net/cas-offinder/[70]) was used to validate the plasmid-targeting sgRNAs used in this study (except sg#2-i and sg#3-i) exhibited no on-target sites on the human genome. The two exceptions were sgRNAs derived from an earlier report testing dCas9/sgRNA binding rules, where it was shown that neither sgRNA's on-target site was at an active gene[71]. For activation of endogenous genes (*IFNG* and mouse *Ifng*, *CCL21*) or Cas9-mediated cleavage (*TP53*), an online tool CRISPR-ERA (http://crispr-era.stanford.edu[72]) was used for sgRNA design[72]. All spacer sequences for sgRNAs are listed in Supplementary Table 2.

**Cell culture and transfection**. 293T (CRL-3216), H1299 (CRL-5803), A549 (CCL-185), LLC (CRL-1642) cells were obtained from ATCC. None of the cell lines used are listed in the ICLAC database. As previously reported, the H1299 and LLC cells showed deficiencies in *p53*, while the A549 cells were confirmed to harbor Kras mutation (G12S). Peripheral blood mononuclear cells (PBMC) were isolated from healthy donors with informed consent. 293T, LLC cells and primary MEFs were cultured in DMEM supplemented with 10% fetal bovine serum (FBS). The same early passage (second) of MEFs were used in the experiments. H1299 and A549 cells were cultured in RPMI-1640 supplemented with 10% FBS and 10 mM HEPES. After isolation via centrifugation through a Ficoll density gradient[73], the PBMCs were cultured in the above RPMI-based medium containing 50 ng/ml human M-CSF (Pepro Tech). These cells were fed with a conditioned medium from p53-tet cells (2× volume over the original culture) to test IFNγ bioactivity. All transfections were carried out using Lipofectamine 3000 according to the manufacturer's instructions. For transfection of the circuits, the vectors encoding dCas9, sgTGT, sgOFF (or AcrIIA4), and the targeted reporter were used at a ratio of 1:1:1:0.5 (or 1:1:1:0.1 for the inhibitable reporter).

**Generation of stable transgenic or gene-knockout/knock-in cell lines**. The transgene-containing lentiviral particles (third-generation) were produced by co-transfection of 293T cells with the lentiviral vector and two packaging plasmids (psPAX2 and pMD2.G) at a ratio of 1:1:1. The virus-containing supernatants were collected at 48 h post-transfection. The filtered (0.45 μm) supernatants were then used to feed the target cells. The p53-null H1299 cells were transduced with the pTRIPZ-p53 virus followed by puromycin selection for 1 week.

The resulting transductants were further diluted to grow cell clones. Different clones were transfected with a P$_{M2}$-EGFP plasmid to test their response to p53. We selected one highly DOX-responsible (p53-tet) clone for propagation and further transfections with synthetic gene circuits. In some experiments, the p53-tet transductants were subjected to the second round of gene introduction with lentiviral particles containing P$_{M2}$-ACRmax-IRES-Ruby. Subsequently, single clones were isolated and tested for DOX-mediated increase of Ruby fluorescence. Two highly responsive clones (P$_{M2}$-Con #13 and P$_{M2}$-ACRmax #16) were selected for further experiments.

To knock out the *TP53* gene in A549 cells, the plasmids for spCas9 and a combination of two targeting sgRNAs (against sites in exon 3 and 4, respectively) were co-transfected. Forty-eight hours later, the transfectants were first enriched by selection in a medium containing both blasticidin and puromycin. After three passages, the cells were plated at a very low density to grow distinctive colonies. The picked colonies were screened by transfection with a $P_{M2}$-EGFP plasmid. The same number (6) of fluorescent (control) or non-fluorescent clones were selected. In the latter clones (p53-null), the mutations at the *TP53* locus were validated by Sanger sequencing.

In another tumor cell system (mouse LLC), the cDNA of p53 was first cloned. Multiple clones were sequenced to determine biallelic mutation patterns. Next, CRISPR/Cas9-mediated WT *p53* knock-in at its endogenous locus in LLC cells was carried out according to an established methodology[74]. Briefly, a genome-targeting sgRNA was designed against a site within the 3′ intron (intron 2) adjacent to the first coding exon (exon 2). The repair template was engineered to contain a *p53* (CDS)-P2A-BSD cassette, led by an 800 bp of intron 1 sequence (left arm) and followed by an 800 bp of genomic sequence just downstream of the cleavage site (right arm). The homolog arms were flanked by two identical sgRNA target sites. This design ensures that co-delivery of corresponding sgRNAs would lead to coordinated genome cleavage and the release of homologous arm flanked template, which in turn would favor precise gene integration assisted by the homology-mediated end-joining mechanism. A promoter-less plasmid (pGL3) was used as the backbone of the repair construct so that the correct recombination events would be enriched in BSD-resistant cells. The gene-targeting was initiated by co-transfecting the LLC cells with plasmids for Cas9, sgRNAs, and the repair template. The cells were selected under BSD for two weeks. The resulting cells were PCR genotyped with primers annealing to the knocked-in sequence and to the surrounding genomic sequences (Supplementary Table 3). The PCR products were additionally sequenced for verifications.

To examine whether an evenly distributed circuit could selectively target the p53-deficient subpopulation, we particularly labeled the parental LLC cells with a mCherry-expressing lentiviral vector. The transduced cells were selected using puromycin. The transductants were validated to be predominantly mCherry⁺. They were used in a co-culture with the unlabeled p53⁺ LLC derivatives.

On the other hand, to test the in vivo effect by our IFNγ-inducing logic circuit, the LLC and their p53⁺ derivatives were respectively introduced with the same circuit. The fluorescence-labeled lentiviral vectors ($LV_{FL}$) containing either the $P_{Suv}$-dCas9 (with mCherry) or tandem of MPH/sgIfn/$P_{M2}$-ACRmax (with EGFP) were constructed. Concentrated viral supernatants from 100 mm of packaging cells were used in a mix to transduce $2 \times 10^5$ cells. Seven days after transduction, some cells were subjected to flow cytometry to examine transduction efficiency.

**Co-culture-based circuit selectivity assay**. The parental LLC cells (mCherry⁺) and the p53⁺ counterparts (mCherry⁻) were mixed 1:1. The $P_{Suv}$-CRISPRa/$P_{M2}$-ACRmax circuit for conditional EGFP activation was transfected to the co-culture. A circuit with a mock inhibitory module ($P_{M2}$-Con) was used as a control. After 24 h, the cells were subjected to flow cytometry analyses.

To test the selectivity by viral vectors-introduced gene circuit to particularly rewire endogenous genes, we subjected the above LLC and p53⁺ co-culture system to circuit delivery via lentiviral vectors. To this end, lentiviral vectors (non-fluorescent, $LV_{NF}$) containing either the $P_{Suv}$-dCas9 or tandem of MPH/sgIfn/$P_{M2}$ACRmax were used to co-transduce the co-cultured cells. Seven days after transduction, the cells were subjected to sorting based on mCherry fluorescence. The RNAs were prepared from the sorted cells for qPCR analyses.

**Animals**. All mice used were of C57/BL6 background and were housed in a humidity- and temperature-controlled, specific pathogen-free facility under a 12:12 h light/dark cycle. The $p53^{+/-}$ mice were originally obtained from the Jackson lab. The genotypes were carried out as described previously[75]. Briefly, the genomic DNA samples were prepared from the toe or tail tips of postnatal mice via overnight digestion with protease K at 55 °C. The samples were next subjected to PCR with a mix of three genotyping primers (see Supplementary Table 3). The expected product sizes for the wild-type and knockout alleles were 450 and 650 bp, respectively. Primary MEFs of different genotypes were obtained from 13.5 to 14.5 days embryos.

**Mouse tumor model**. The parental LLC cells and their p53⁺ derivatives were introduced with the conditional immune-rewiring circuits via lentiviral vectors in vitro. Judged by fluorescent markers, the efficiency of complete circuit introduction (two vectors in combination) is about 6%. Totally, $2 \times 10^6$ of such transduced cells (without fluorescent selection) were inoculated subcutaneously to the flanks of C57BL/6J mice (6–8 weeks). Once the tumors are palpable, their sizes were measured every 2 days using a vernier caliper. The tumor volume was calculated using the formula: volume $= x \times y^2/2$ (*x*: tumor length; *y*: tumor width).

To analyze the immune compartment in the tumors, the samples were prepared as described before[76]. Briefly, the tumor was cut into small pieces and was further dissociated with the help of enzymatic digestion (collagenase I [170 mg/L], collagenase II [56 mg/L], and DNase I [25 mg/L]) for 30 min at 37 °C. The filtered (via 80 μm nylon mesh) cell suspension was next subjected to red blood cell removal using the ACK buffer. The cells were washed with PBS twice before proceeding to antibody staining.

**Flow cytometry**. The adherent cells were either trypsinized (with GFP reporter) or non-enzymatically dissociated (PBS containing 0.5% EDTA). For cultured PBMC, the dissociated and suspended cells were combined. The cell suspensions were subsequently stained with fluorochrome-labeled antibodies for 1 h on ice (at 1:100 dilution for all antibodies). The conjugated antibodies were all purchased from BioLegend. They include those against human CD45 (APC, 304012), CD11b (FITC, 101205), HLA-ABC (FITC, 311404), and HLA-DR (PE, 307606), together with those against mouse CD45 (APC/Cy7, 103116), CD4 (FITC, 100406), CD8a (BV421, 100737), and CD3ε (APC, 100311). Flow cytometry analyses were carried out using BD LSRFortessa or BD FACS Calibur. Data were analyzed using the FlowJo software.

**Real-time quantitative PCR, Western blotting, and luciferase analyses**. The RT-qPCR and Western blotting (immunoblotting, IB) were performed as previously described[77]. A brief summary is provided as the following. For RT-qPCR, total RNA was extracted from the cells or tumors using RNAiso Plus (Takara, #9109), and then subjected to reverse transcription with the aid of HiScript Q-RT SuperMix (Vazyme, R123-01). The samples were then aliquoted for real-time PCR using AceQ qPCR SYBR Green Master Mix (Vazyme, Q141-AA). The levels of GAPDH were used as internal controls for normalization. For IB, cells were harvested using RIPA buffer (50 mM Tris-HCl, pH 7.4, 150 mM NaCl, 1% Triton X-100, 1% sodium deoxycholate, 0.1% SDS) supplemented with 1 mM of NaF, 1 mM of $NaVO_3$, and the protease inhibitor cocktail [1:50 dilution, Sigma, P8340]). Samples were resolved on SDS-PAGE gels and were transferred to PVDF membranes. Following successive incubation with primary/secondary antibodies, the immunoreactivity was determined by using a chemiluminescent substrate (Tanon ECL, #180-5001). The primary antibodies for spCas9 (A01935-40) and actin (A00730) were from GenScript. The primary antibodies for GAPDH (SC32233) and p53 (SC126) were from Santa Cruz. Other various vendors supplied the primary antibodies for EGFP (Abclonal, AE012), STAT1 (Sangon Biotech, D155186), pSTAT1 (Cell Signaling Technology, 7649S), and Flag (Sigma, F1804). The primary antibodies were used at 1:1000 dilution, except for spCas9 (1:500), p53 (1:500), and EGFP (1:2000). The raw data for the presented blotting results can be found in the Source Data file. The primers for RT-qPCR are listed in Supplementary Table 4. In some experiments to measure promoter activities, cells in triplicates were co-transfected with separate plasmids encoding inducible promoter-driven firefly luciferase and CMV-dependent renilla luciferase at a 50:1 ratio. After 24–48 h of treatment, cells were lysed and the reporter activities were determined by a dual-luciferase assay kit from Promega. The relative inducible promoter-dependent transcriptional activities were presented as the ratio of Fluc/Rluc (±SD).

**RNAseq analyses**. The LLC and their p53 knocked-in derivatives (p53⁺) were transfected with the $P_{Suv}$/$P_{M2}$ AND-NOT circuit (with ACRmax) for 48 h. Total RNA was isolated and subjected to commercial RNAseq services (Annoroad Gene Tech., Beijing). The mRNAs were enriched, purified, and fragmented to construct DNA libraries for the Illumina platform. The libraries were sequenced using Strategy PE150. The clean reads were mapped against Mus_musculus.GRCm38 genome assembly. Gene expression value was quantitatively estimated through Fragments per Kilobase per Million Mapped Fragments. The Principal Component analyses were performed. Genes with significantly differential expression between groups were determined by DESeq2, using a cutoff of Fold change (FC) ≥ 4 and adjusted $P < 0.05$. GO enrichment analyses were performed using the PANTHER classification system[78].

**MTT cell viability assays**. The transfected cells were fed with fresh media containing Thiazolyl Blue Tetrazolium Bromide (Yeasen) at a final concentration of 0.5 mg/ml. Following 4 h of incubation, the cell layer was added with 500 μl DMSO to extract the formazan product. The absorbances of the extracts were determined in a 96-well plate at 490 nm (690 nm as a reference wavelength).

**Statistical analyses**. All data presented in this study are derived from at least two independent experiments. Representative Western blot results are presented. Sometimes for demonstration purposes, intensities of bands were determined by ImageJ and were normalized (as indicated below the panels of interest). For quantitative data, the average values and variances from biological replicates (or independent experiments) were presented. No statistical method was used to predetermine sample size. No data were excluded from the analyses. Mice of the same age and sex were randomly divided into cohorts for parallel inoculation with different groups of tumor cells. Randomization was not applied to in vitro cell line experiments. Based on the objective nature of the measurements, blinding was not applied in the present study. The error bars denote SEM or SD ($n \geq 3$), or sometimes the data range (when $n = 2$), as indicated in figure legends. $P$ values were determined by Student *t*-tests.

**Reporting summary**. Further information on research design is available in the Nature Research Reporting Summary linked to this article.

## Data availability

High-throughput RNAseq data (raw and processed) are deposited at the NCBI GEO database with the accession number of GSE179814: The reference mouse genome assembly Mus_musculus.GRCm38 is an openly accessible resource (https://www.ncbi.nlm.nih.gov/assembly/GCF_000001635.20/). Source data are provided with this paper.

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

## Acknowledgements

This work is supported by grants from the National Key Research and Development Program of China (2019YFA0802800 and 2021YFF1000704 to J.L.), the National Natural Science Foundation of China (81771784 [to Q.L.], 32101169 [to Q.M.], 31771574 [to J.L.]) and the fellowship of China Postdoctoral Science Foundation (2021M701680 [to Y.W.]).

## Author contributions

Y.W., G.Z., Q.M., P.G., and S.H. performed the experiments. S.H. analyzed the RNAseq data. Y.W., G.Z., and Q.M. analyzed most experimental results. Q.L, L.S., G.L., X.H., and J.L. supervised and designed the experiments. Y.W., G.L., X.H., and J.L. wrote and edited the paper.

## Competing interests

The authors declare no competing interests.
