## [Peer Review File · Nature Communications]

Reviewers' Comments:

Reviewer #1:

Remarks to the Author:

The work presented here represents a strategy in which gene expression can be regulated epigenetically under specific conditions. Demonstrated here is the proof of concept of a "logic" circuit in which the presence/absence of key tumorigenic transcription factors are able to activate/inactivate epigenetic activities. The presented work is logically developed and works as expected in-vitro. The thoroughness demonstrated through their design and implementation of several logic gates looks to be beneficial in the future development of genetic therapies. Of note, the use of anti CRISPR molecules for repression far exceeds the ability of simple promoter-based techniques, eliciting a far greater "off switch" effect. The development and simplification of this system into a single unit looks to improve the implementation of this technology. Overall, the design and execution of this system looks promising in providing a new toolset in which epigenetic CRISPR activities can be implemented, and specifically targeted to cells of interest. Furthermore, it looks as if this system could be easily remodelled to respond to a variety of inputs, thus not limiting its implementation to p53 repressed/expressing cells. The authors have been diligent in testing the efficacy of this system, with gene expression and protein expression checked at every stage of the development. In the final demonstration they were able to show effective increases in IFN γ expression, and the subsequent expected gene expression changes associated with increased IFN γ .

Major comments:

One very significant drawback is the lack of in-vivo validation of this system. While they have successfully demonstrated its ability in cell culture, without in-vivo testing it is difficult to determine how useful this technology is either for diagnosis or for treatment. Especially as the primary rationale is the selective and differential ability of this system to regulate gene expression only in cells with the correct conditions (eg. Low p53). One validation that would potentially help prove the efficacy of this system would be testing in a co-culture system where cell harbouring different p53 levels could be assayed?

This work could be considerably stronger if an in vivo application could be demonstrated. For example, would T cells or different immune populations be engaged by these technologies? Would these technologies enable specific recruitment of immune effectors in mouse models? Would they enable sensitization/response to immune-check point blockades?

Additionally, the inclusion of a schematic detailing the whole system would be greatly beneficial, especially as the authors have discussed several iterations of the system throughout its development, making the whole document harder to understand.

Minor issues: The language used throughout the document at times is confusing, making understanding the work more difficult than would be desirable. Prime examples of this are in the abstract, which would benefit from simplification, especially as this is the first thing anyone will read.

More language issues are found throughout e.g.

86 Synthetic gene circuits hold considerable

87 potential to enable tumor-specific delivery of immunological interventions. Indeed, as

88 the immune regulators act in non-cell-autonomous manners, they are suited as

89 functional outputs for gene-based, partially delivering systems 11.

At times the results read as if discussing what future work/developments could be done, when in fact, the work has already been done.

The whole document would greatly benefit from proofing, with general comprehension a priority. Please note qRT-PCR data should at least contain biological triplicates.

The discussion should at least mention some recent works on the utilization of CRISPRa for the immune modulation as there are already some published reports utilizing these systems in PUBMED, as well as recent reviews in the topic.

Reviewer #2:

Remarks to the Author:

The manuscript "Precise tumour immune rewiring via synthetic CRISPRa circuits gated by

concurrent gain/loss of transcription factors" by Wang et al. presents an innovative approach to selectively enable immunotherapy in poorly immunogenic tumours.

The synthetic biology tools described in this paper, combined with the originality of the idea, are of great interest for both the synthetic and cancer biology communities. If fully developed, such tools could help tumour specific targeting, encompassing immunoregulatory genes re-activation (as described by the authors) but also cell-specific gene editing.

The article is well written, and the results are overall clearly presented. Experiments are generally well designed, although additional controls could have occasionally been added to clarify certain aspects of the conclusions drawn.

I have however to raise some criticism, mainly about the structure of the manuscript, some unnecessary speculations occurring here and there and an apparent lack of consequentiality between the first and second AND/NOT gate designs.

Figure-specific comments are added at the end of this letter, while general thoughts on the manuscript and possible improvements are suggested below.

General comments

The article is mainly constructed around building and optimization of AND/NOT gates strategy to ensure CRISPRa activation in presence of a tumour-specific transcription factor (active in tumours) and lacking a second transcription factor (absent in tumours). The first three figures (and relative supplementary) revolve around optimization of AND/NOT gates involving polII promoters driving the expression of inhibitory sgRNAs (NOT gate). Although the experiments here described are elegant and carefully designed, they fall short in terms of specific activation within the tumour cells. In all the experiments, leaky expression of dCas9 and one of more sgRNAs weaken the strategy because of leaky expression of target genes.

In view of the final goal (i.e., tumour-specific targeting) this is a major obstacle, and I feel the authors should describe less triumphantly the outcomes of the experiments shown in the first four figures. While they are encouraging (reduced expression of dEGFP in presence of NOT gating) they would certainly not be enough to develop a therapeutic strategy, as these constructs would inevitably lead to mild expression of target genes in non-target (normal) cells, with unpredictable effects.

I encourage the authors to openly discuss the limitations of these polII expression systems in the discussion (where only the final approach is discussed). Additionally, as the authors correctly point out, the AND/NOT gate logic is weakened when using activating and repressing modules both based on CRISPRa/i (using the same protein effector for both activation and repression compartments of the circuit).

Figure 5 contains the most convincing experiments of the whole manuscript. However, here the design strategy is drastically changed, the polII-sgRNA expression and NOT-gating are abandoned (without sufficient explanations in my opinion) and the AND/NOT gate is built around a tumour specific transcription factor promoter (driving the expression of CRISPRa elements) and a NOT gate using the CRISPR inhibitor AcrIIA4 under a non-tumour only specific promoter (e.g. pMd, sensitive to p53). This approach greatly (if not completely) reduces leaky target genes activation in normal cells, as demonstrated by both introduction of p53 in p53 null cells and its opposite, across different tumour cell models. The results provide convincing evidence that the new design is superior to the previous ones and that it selectively activates target genes only within tumour (e.g. p53 negative) cells.

In absence of additional experiments, however, I am not convinced that the ACRmax provides a substantial improvement over the AcrII4 Cas9 inhibitor. No experiments (except for a superior expression level of the coupled mRuby fluorescent proteins) are provided to justify the use of ACRmax.

I am also not fully convinced that the AND/NOT logic of the final construct contributes to the overall success of the experiments shown. In my opinion, the NOT component (now rewired around the AcrII4) is now extremely powerful and perhaps sufficient for the experimental outcome. In this context, no experiments are provided to show the usefulness of the AND branch of the circuit.

For instance, can the author test whether they have a similar target genes activation when Psuv is replaced with a constitutive promoter (e.g. CMV) to drive dCas9 expression? Would a weak promoter (e.g. SV40) work even better by enabling lower expression of dCas9 and a tighter NOT

control by pMd ACRmax, without the need of a cell-specific promoter? I think that these experiments are particularly needed in any of the models with p53+/- conditions, to provide clear insights into the usefulness of the "AND" branch of the circuit.

Finally, I suggest mentioning in the discussion possible delivery-related issues.

Figure-specific comments:

Figure S2b

Can the author comment why the polII-driven sgCUT expression cassette has a much lower expression efficiency as compared to the U6-driven cassette? Also, the basal eGFP expression level is unexpectedly high. A lack in sgRNA efficiency expression under a polII promoter can seriously hinder the efficacy of the subsequent AND/NOT gate (presented later in the manuscript), leading to target genes activation in non-cancer cells (e.g. leaky expression in presence of p53).

The plots should be presented with mRuby on the Y-axis, and eGFP on the X-axis. Are the actual plots showing populations of cells already gated for mRuby+ cells? If so, this is not specified in the legend. Additionally, a CMV HHRibo-sgLuc-HDRBRibo control cassette is missing.

Figure S2c

How does sgTGTi behave when expressed from a U6 promoter? This control is required to establish the robustness of the system. The construct 5xISRE dCas9-KRAB appears to be leaky. A mild expression of Fla-dC-KRAB is detectable in basal conditions and, while the IFNa treatment induces a substantial upregulation of the target gene, the basal expression levels are, in my opinion, enough to drastically reduce the expression of dEGFP (please compare eGFP lanes with IFNa 0 in SgCon-i and sgTGT-i transfections). This design leads to basal dEGFP downregulation regardless the IFNa input, due to leaky dCas9 expression.

Figure 2b

The leaky expression is once again interfering with the overall design logic, undermining the final outcome. Basal dCas9 and (consequently) eGFP expression can be appreciated in IFNa treatment (western blot and IFN- cells).

Technical comment: Scalebar and type of experiment of the figure inset are not described.

Figure S2d

Given the residual expression of dCas9-KRAB in sgISRE-i +INFa, the lack of reduction of eGFP is somewhat surprising (while similar levels are enough to induce eGFP downregulation in FigS2c). It seems that the sgOFF control is tighter on the sgTGTi and this negative regulation might be sufficient to exert a similar effect. Given that PolII sgRNA are suboptimally expressed as deduced by FigS2b, they might represent the rate-limiting factor of the gate logic, rather than the dCas9-KRAB module. Can the authors comment on this and add experimental conditions (controls) in which the P1 promoter on the dCas9-KRAB is constitutively combined with 5xISRE sgTGTi?

Figure S2e

It is not clear if the sgOFF sequence is the same of FigS2d or it's a different one. Again, upon target optimization, it seems that, although it exerts its function, sg#2-i is not as efficient as its U6 driven counterpart (FigS2d). A mild downregulation of eGFP can still be appreciated, and dCas9 downregulation is not as tight as in FigS2d (in which was already not complete). On the other hand sg#1-i is seemingly not working. Despite the elegant logic, it seems to me that, in view of the final application of the system, the polII dampened sgRNA expression efficiency must be resolved.

Figure s2f

The unique nomenclature of the sgRNA OFF is again confusing, as it appears to be referred to another sequence if I understood properly. Can the author comment on why they didn't insist on using sgISRE-i in Figure S2d?

Figure 2c

Compared to the experiment shown in FigS2f, with the same sgRNAs, the introduction of the MS2-p65-HSF1 module under the control of the 5xISRE element, appears to significantly enhance dCas9 leaky expression compared to a constitutively expressed cassette (FigS2f). This is

counterintuitive as it is suggestive of crosstalk between sgOFF (which should be deprived of MS2 stem-loops) and the MS2-p65-HSF1 module. Additionally, leaky expression in absence of INF α is not completely abrogated.

Figure 2e, f

Although the system is working, leaky expression of Cas9 and sgTGTa sgRNA contribute to the AND/NOT logic to not be completely bullet-proof. In a real-life scenario, the system would induce leaky expression of target genes in normal cells too.

Figure S4a

Western blot on mRuby is not a direct proof of increased expression levels of ACRmax compared to AcrIIA4. Also, the superior efficiency of ACRmax is not strongly supported by additional data (e.g. similar eGFP induction in Fig S4b). In absence of additional data, the choice of ACRmax is somewhat arbitrary, although the authors should not stress its superior efficiency based on codon optimization and mRuby western blot only. A western blot of eGFP levels in Figure S4b experiment could be performed to clarify this point.

Below are our point-by-point replies:

Reviewer #1 (Remarks to the Author):

The work presented here represents a strategy in which gene expression can be regulated epigenetically under specific conditions. Demonstrated here is the proof of concept of a “logic” circuit in which the presence/absence of key tumorigenic transcription factors are able to activate/inactivate epigenetic activities. The presented work is logically developed and works as expected in-vitro. The thoroughness demonstrated through their design and implementation of several logic gates looks to be beneficial in the future development of genetic therapies.

Of note, the use of anti CRISPR molecules for repression far exceeds the ability of simple promoter-based techniques, eliciting a far greater “off switch” effect. The development and simplification of this system into a single unit looks to improve the implementation of this technology. Overall, the design and execution of this system looks promising in providing a new toolset in which epigenetic CRISPR activities can be implemented, and specifically targeted to cells of interest. Furthermore, it looks as if this system could be easily remodelled to respond to a variety of inputs, thus not limiting its implementation to p53 repressed/expressing cells.

The authors have been diligent in testing the efficacy of this system, with gene expression and protein expression checked at every stage of the development. In the final demonstration they were able to show effective increases in IFN γ expression, and the subsequent expected gene expression changes associated with increased IFN γ .

Major comments:

One very significant drawback is the lack of in-vivo validation of this system. While they have successfully demonstrated its ability in cell culture, without in-vivo testing it is difficult to determine how useful this technology is either for diagnosis or for treatment. Especially as the primary rationale is the selective and differential ability of this system to regulate gene expression only in cells with the correct conditions (eg. Low p53). One validation that would potentially help prove the efficacy of this system would be testing in a co-culture system where cell harbouring different p53 levels could be assayed?

[Authors' reply]

We thank the reviewer very much for pointing out some strengths of our work, and for also suggesting the directions to improve the study. We agree that to further test the performance of our tumor-activatable synthetic circuit in models that mimics the therapeutic context would be informative.

As suggested by the reviewer, we established a co-culture of LLC cells (p53-deficient, labeled with mCherry) and their p53⁺ knock-in derivatives, respectively representing the targeted and non-targeted subpopulations. When the mixed cells were transfected with the P_{Suv}-CRISPRa/P_{M2}-ACRmax circuit for EGFP activation, the subpopulation of LLC cells (mCherry⁺) showed a markedly greater level of EGFP induction than the p53⁺ subpopulation (current Fig. 5d, line 390-404).

Currently, therapeutic tumor targeting by a gene circuit is likely to involve viral vector-mediated delivery methods. As a proof-of-principle for viral circuit delivery, we also constructed a mix of two lentiviral vectors incorporating the P_{Suv}/P_{M2}-directed IFN γ -inducing circuit. Similar to the design above, a co-culture of LLC and their p53⁺ derivatives were transduced with these lentiviral vectors. Consistently, the viral vector-incorporated circuit also exhibited a far greater activity to drive *Ifng* induction in the subpopulation of LLC cells (mCherry⁺) than in their p53⁺ counterparts (current Supplementary Fig. 6d). Collectively, these *in vitro* experiments demonstrate the ability by our circuit to specify the p53-null subpopulation upon an evenly distributed delivery (line 432-439).

We realize that the AAV vectors may be more therapeutically relevant. Two serotypes of fluorescent protein-labeled AAV vectors (AAV5 and AAV9, vector-only), with reported tropism for certain lung cancer cells and on lung epithelium in mouse models^{1, 2}, were obtained commercially (Genechem Inc, Shanghai). Unfortunately, at even a high MOI of 1000, neither of these AAV vectors could efficiently transduce LLC cells (efficiencies at ~1% and ~7%, respectively). It can be expected that such an issue with transduction efficiency would be further exacerbated when delivering our circuit (a mix of 2 vectors with large payloads). Consequently, the AAV approach was not further pursued.

This work could be considerably stronger if an in vivo application could be demonstrated. For example, would T cells or different immune populations be engaged by these technologies? Would these technologies enable specific recruitment of immune effectors in mouse models? Would they enable sensitization/response to immune-check point blockades?

[Authors' reply]

We thank the reviewer very much for suggesting experiments to test the therapeutic activities by our tumor-selective, immune-activating logic circuit *in vivo*. In LLC-based transplantable tumor models, a comparison of circuit-driven effects in the parental LLC tumors and their p53⁺ counterparts (knocked-in) would suggest the therapeutic potential and targeting specificity by this tool in an *in vivo* context. Indeed, transplantation of the circuit-transduced LLC cells (parental or p53⁺) would serve as a convenient strategy to provide important insights (current Fig. 6c, d and Supplementary Fig. 6e-h, line 440-474).

To this end, we further developed a mix of two lentiviral vectors with fluorescent labels (LV_{FL}) to package the P_{Suv}-CRISPRa-*Ifng*/P_{M2}-ACRmax circuit. The fluorescent labels would enable convenient assessment of transduction efficiencies. The LLC cells and their p53⁺ derivatives were respectively transduced *in vitro*, and equivalent efficiencies of full-circuit transduction were demonstrated (current Supplementary Fig. 6e). These cells were next transplanted s.c. to C57/BL6 mice. Importantly, the *in vivo* progression of the LLC, but not the p53⁺ tumors, was notably inhibited by the circuit (current Fig. 6c). In addition, the circuit-introduced LLC tumors (but not their p53⁺ counterparts) showed signs of enhanced cytotoxic T cell activities in association with activation of the IFN γ axis and an immune-stimulatory program (current Fig. 6d). It is

also worth noting that these *in vivo* effects were achieved with full circuit introduction to only a portion of the LLC cells (see current Supplementary Fig. 6e). These results provide support for the therapeutic potential and specificity of our tumor immune-rewiring gene circuit.

Interestingly, consistent with a paradoxical role of IFN γ in the induction of certain immunosuppressive signals, the circuit-introduced LLC tumors also showed an increase in the levels of PD-L1 mRNA (see current Fig. 6d). Such observation suggests a potential strategy to combine the application of our tumor-specific immune-rewiring circuit with the existing checkpoint inhibition modalities. This prospect is added to the Discussion (line 574-579). Overall, we are very grateful to the reviewer's suggestion on the *in vivo* testing of our circuit, which looks to be an exciting direction for uncovering further insights.

Due to the inefficiency of transducing LLC cells by the more therapeutically relevant AAV vectors, as described earlier, the present study did not explore *in vivo* circuit delivery in a treatment regimen. However, to move forward, we realize the importance of adapting our circuit for treatment models through the use of clinically relevant viral vectors, or other cutting-edge delivering technologies. Such account is now added to the discussion (line 584-589).

Additionally, the inclusion of a schematic detailing the whole system would be greatly beneficial, especially as the authors have discussed several iterations of the system throughout its development, making the whole document harder to understand.

[Authors' reply]

We thank the reviewer very much for this helpful suggestion. For improvements in clarity, we placed two related schematics to the revised figures, corresponding to a brief description of the present work within the Introduction (line 80-84). The first schematic is for the overall circuit design (current Fig. 1a), while the second is for the overall flow of the presented experiments (current Supplementary Fig. 1a). An older schematic in the original manuscript is removed (original Fig. 2a).

In the current Fig. 1a, two sequential versions of the circuit are denoted (v1 and v2), corresponding to the major developments in the NOT gate. They respectively feature a TF2-sensing inhibitory module (pink) against either the expression (v1) or activity (v2) of the dCas9 effector in the activation module (grey).

Minor issues: The language used throughout the document at times is confusing, making understanding the work more difficult than would be desirable. Prime examples of this are in the abstract, which would benefit from simplification, especially as this is the first thing anyone will read.

[Authors' reply]

We apologize for the language clarity issue. The abstract has been simplified to facilitate understanding.

*More language issues are found throughout e.g.
86 Synthetic gene circuits hold considerable*

87 potential to enable tumor-specific delivery of immunological interventions. Indeed,
as

88 the immune regulators act in non-cell-autonomous manners, they are suited as
89 functional outputs for gene-based, partially delivering systems 11.

At times the results read as if discussing what future work/developments could be done,
when in fact, the work has already been done.

The whole document would greatly benefit from proofing, with general comprehension
a priority.

[Authors' reply]

Again, we sincerely apologize for the language and clarity issues. The concerned
confusing text has been corrected. The language throughout the manuscript has been
edited for better understanding by the readers.

Please note qRT-PCR data should at least contain biological triplicates.

[Authors' reply]

We thank the reviewer for this critical reminder. We re-examined our qPCR data,
and performed necessary replications. All qPCR results in the revised manuscript now
convey information from at least biological triplicates.

*The discussion should at least mention some recent works on the utilization of CRISPRa
for the immune modulation as there are already some published reports utilizing these
systems in PUBMED, as well as recent reviews in the topic.*

[Authors' reply]

We thank the reviewer for this important suggestion. We have added description of
some recent works that harnessed CRISPRa for immune activation (line 481-483).

Reviewer #2 (Remarks to the Author):

*The manuscript "Precise tumour immune rewiring via synthetic CRISPRa circuits
gated by concurrent gain/loss of transcription factors" by Wang et al. presents an
innovative approach to selectively enable immunotherapy in poorly immunogenic
tumours.*

*The synthetic biology tools described in this paper, combined with the originality of the
idea, are of great interest for both the synthetic and cancer biology communities. If fully
developed, such tools could help tumour specific targeting, encompassing
immunoregulatory genes re-activation (as described by the authors) but also cell-
specific gene editing.*

*The article is well written, and the results are overall clearly presented. Experiments
are generally well designed, although additional controls could have occasionally been
added to clarify certain aspects of the conclusions drawn.*

*I have however to raise some criticism, mainly about the structure of the manuscript,
some unnecessary speculations occurring here and there and an apparent lack of*

consequentiality between the first and second AND/NOT gate designs.

Figure-specific comments are added at the end of this letter, while general thoughts on the manuscript and possible improvements are suggested below.

[Authors' reply]

We thank the reviewer for an overall positive appraisal, and for making suggestions to improve our work. In response to the comments regarding the structure of our manuscript, we apologize for having not smoothly connected the two sections in the original manuscript, where the experiments described involve different versions of the NOT gates.

In the revised manuscript, we placed more emphases on highlighting the staged progression of circuit development. Indeed, we openly stated the incomplete effects by the inhibitory sgRNA (sgOFF)-based NOT gates through the “Results” section (line 225-228, 237-240, 244-247, 272-274). In addition, at the junction between the first and second section, some comments were made to stress the need for a better performing NOT-gating strategy (line 288-292, and 293-298).

We would also like to point out that, for the second NOT gate, some additional *in vivo* testing in a mouse model were performed. The new data are now placed in the current Fig. 6 (line 440-474). Taken together, the revised manuscript has made efforts on presenting the rationales for the two-stage circuit development, and has further extended the exploration of the better performing, AcrIIA4-containing circuit.

General comments

The article is mainly constructed around building and optimization of AND/NOT gates strategy to ensure CRISPRa activation in presence of a tumour-specific transcription factor (active in tumours) and lacking a second transcription factor (absent in tumours). The first three figures (and relative supplementary) revolve around optimization of AND/NOT gates involving polIII promoters driving the expression of inhibitory sgRNAs (NOT gate). Although the experiments here described are elegant and carefully designed, they fall short in terms of specific activation within the tumour cells. In all the experiments, leaky expression of dCas9 and one of more sgRNAs weaken the strategy because of leaky expression of target genes.

In view of the final goal (i.e., tumour-specific targeting) this is a major obstacle, and I feel the authors should describe less triumphantly the outcomes of the experiments shown in the first four figures. While they are encouraging (reduced expression of dEGFP in presence of NOT gating) they would certainly not be enough to develop a therapeutic strategy, as these constructs would inevitably lead to mild expression of target genes in non-target (normal) cells, with unpredictable effects.

I encourage the authors to openly discuss the limitations of these polIII expression systems in the discussion (where only the final approach is discussed). Additionally, as the authors correctly point out, the AND/NOT gate logic is weakened when using activating and repressing modules both based on CRISPRa/i (using the same protein effector for both activation and repression compartments of the circuit).

[Authors' reply]

We thank the reviewer very much for carefully examining our data and for making

these insightful comments. As mentioned in our response to the last point, we modified the text to openly convey the incomplete NOT gate performance with the sgOFF strategy in the Results section. Indeed, such would serve as a reference to highlight the substantial improvement by our second NOT gate.

In addition, the reviewer's comments also inspired us to extend some discussions about the limitations and the potential advantages with the sgOFF system (line 517-526). We believe such open discussions would be of interest to the synthetic biology community and beyond.

Figure 5 contains the most convincing experiments of the whole manuscript. However, here the design strategy is drastically changed, the polII-sgRNA expression and NOT-gating are abandoned (without sufficient explanations in my opinion) and the AND/NOT gate is built around a tumour specific transcription factor promoter (driving the expression of CRISPRa elements) and a NOT gate using the CRISPR inhibitor AcrIIA4 under a non-tumour only specific promoter (e.g. pMd, sensitive to p53). This approach greatly (if not completely) reduces leaky target genes activation in normal cells, as demonstrated by both introduction of p53 in p53 null cells and its opposite, across different tumour cell models. The results provide convincing evidence that the new design is superior to the previous ones and that it selectively activates target genes only within tumour (e.g. p53 negative) cells.

In absence of additional experiments, however, I am not convinced that the ACRmax provides a substantial improvement over the AcrII4 Cas9 inhibitor. No experiments (except for a superior expression level of the coupled mRuby fluorescent proteins) are provided to justify the use of ACRmax.

[Authors' reply]

We thank the reviewer for acknowledging the superior performance by the second AcrIIA4-based NOT gate. We also appreciate the request for a more indicative control to justify our adoption of the codon-optimized ACRmax. As per reviewer's suggestion, we further compared the CRISPRa-inhibiting effects by an AcrIIA4 in original codons and its counterpart in human optimized codons (ACRmax) via a Western blot for the levels of the targeted EGFP reporter (current Fig. 4a, line 306-309). Both dCas9 and AcrIIA4 forms were expressed under CMV promoters. Although the AcrIIA4 in bacterial codons showed considerable activities in inhibiting CRISPRa function, the ACRmax was visibly even more effective. To increase the likelihood by our inhibitory module to operate under basal TF (p53) activities, we chose to use the more potent ACRmax through the rest of the study. For simplification, the data for some indirect evidence [on the levels of a downstream IRES-mRuby] (IB in Fig. S4a of the original manuscript), as well as a less quantitative EGFP immunofluorescence panel were removed (Fig. S4b in the original manuscript).

I am also not fully convinced that the AND/NOT logic of the final construct contributes to the overall success of the experiments shown. In my opinion, the NOT component

(now rewired around the AcrII4) is now extremely powerful and perhaps sufficient for the experimental outcome. In this context, no experiments are provided to show the usefulness of the AND branch of the circuit.

For instance, can the author test whether they have a similar target genes activation when Psuv is replaced with a constitutive promoter (e.g. CMV) to drive dCas9 expression?

[Authors' reply]

We thank the reviewer for these comments, and for this question. We concur with the reviewer that the ACRmax-driven NOT gate is very powerful. Indeed, most of our experiments in Fig. 4 involved the use of a constitutive CRISPRa (CMV-dCas9). Here, in a variety of cell lines (A549, MCF-7) and MEFs, P_{M2}-ACRmax driven by endogenous p53 at basal condition can efficiently inhibit EGFP reporter activation by this potent CRISPRa (Fig. 4c-f and S4i of the original manuscript, currently as Fig. 4d-f and Supplementary Fig. 4h). The H1299 (p53-null), p53-knockout A549 and p53^{+/-} MEFs were used as respectively controls. We modified a segment in the Results to better highlight our application of CMV-driven CRISPRa in these experiments (line 356-359). Some related comments are also added to the Discussion to extend such a point (line 534-537).

Would a weak promoter (e.g. SV40) work even better by enabling lower expression of dCas9 and a tighter NOT control by pMd ACRmax, without the need of a cell-specific promoter? I think that these experiments are particularly needed in any of the models with p53^{+/-} conditions, to provide clear insights into the usefulness of the "AND" branch of the circuit.

[Authors' reply]

Further evidence on the stringency of the P_{M2}-ACRmax NOT gate:

We thank the reviewer for this discussion. The suggestion that tuning the strength of dCas9 expression may allow differentiation of WT and p53^{+/-} is very intriguing. However, since the ACRmax module is very potent, the P_{M2}-ACRmax may only allow determination of p53 status in an *all-or-none* fashion.

Indeed, we now add evidence that even a single allele of p53 (p53^{+/-}) in MEFs sufficed to signal P_{M2}-ACRmax into strongly inhibiting a CRISPRa output dependent on SV40-dCas9 (current Supplementary Fig. 4i). The use of CMV-dCas9 presented similar results (not shown). Corresponding text has been added (line 350-353).

Such a potent activity by ACRmax would make a p53-null (or very low) state as a requirement for a circuit output by CRISPRa actuator. Although a circuit adopting a stringent p53 NOT gate would have the limitation to require severe p53 deficiency for output production (thus unable to target some p53-retaining tumors, or p53 heterozygosity), it would feature a lower risk for undesired activation in normal cells, which is a key goal for the current development of therapeutic circuits. Indeed, we suspect that to set an actuator threshold by applying the P_{M2}-dependent inhibitory module for classification of a p53^{+/-} status may bring significant ambiguity, as p53 activities in different cells are likely to vary considerably^{3,4}. Related discussion has

been added (line 537-541).

The importance of the onco-TF-driven branch of the AND-NOT gate:

On the other hand, it is reasonable to assume that the use of a “tumor-specific” promoter for CRISPRa component may readily install a first layer of restriction on the circuit output. To verify this point, we performed additional experiments. We used a constitutive SV40 promoter or a tumor-specific P_{Suv} promoter to drive dCas9 expression. Subsequently, different CRISPRa activities in human (H1299) and mouse (LLC) tumor cells, as well as in normal mouse embryonic fibroblasts (MEFs) were compared (current Supplementary Fig. 1c). In reference to the activity profile of SV40 promoter (constitutive), the P_{Suv} showed apparently higher activities to drive dCas9 expression in tumors cells than in MEFs. Importantly, the CRISPRa activities (reported by EGFP levels) in different groups closely correlated with those for dCas9 expression. These results provide evidence that controlled dCas9 expression by the use of tumor-specific promoters places a first level of restriction on the CRISPRa outputs. Corresponding text has been added (line 101-116).

Therefore, in our overall design of the AND-NOT logic circuit, the tumor-selective promoter for dCas9 would provide one level of control for CRISPRa activity (Input-1), which is additionally gated by the highly stringent p53-responsive ACRmax module (the NOT gate) at the same time (Input-2). As some normal cells in a tissue context may possibly feature low p53 activities insufficient to engage the P_{M2} -ACRmax module, the other layer of restriction for CRISPRa activity set by the tumor selective promoter would further reduce the risk of targeting such normal cells. In this regard, the CRISPRa-introduced MEFs (without an ACRmax module, current Supplementary Fig. 1c) could be viewed as an extreme scenario, where an P_{M2} -ACRmax was hypothetically non-operative. Therein, the application of a “tumor-specific” P_{Suv} -dCas9 module would have served to limit unwanted CRISPRa activities in these normal cells. Related discussions are currently positioned at line 547-551.

Moreover, as we had discussed in the previous version (line 397-402 in the original manuscript), although tumor recognition by our circuit requires p53-deficiency, its usefulness may be extended to individuals bearing heterogeneous lesions with mixed p53 status. Herein, the initial targeting of the p53-null subpopulation may stimulate an overall anti-tumor immunity for all tumors (line 597-601).

Finally, I suggest mentioning in the discussion possible delivery-related issues.

[Authors’ reply]

We thank the reviewer for this suggestion. Texts on delivery-related issues have been added to the discussion (line 584-589).

Figure-specific comments:

Figure S2b

Can the author comment why the polIII-driven sgCUT expression cassette has a much lower expression efficiency as compared to the U6-driven cassette? Also, the basal

eGFP expression level is unexpectedly high. A lack in sgRNA efficiency expression under a polII promoter can seriously hinder the efficacy of the subsequent AND/NOT gate (presented later in the manuscript), leading to target genes activation in non-cancer cells (e.g. leaky expression in presence of p53).

The plots should be presented with mRuby on the Y-axis, and eGFP on the X-axis. Are the actual plots showing populations of cells already gated for mRuby+ cells? If so, this is not specified in the legend. Additionally, a CMV HHRibo-sgLuc-HDRBRibo control cassette is missing.

[Authors' reply]

We thank the reviewer for these questions. Indeed, although Cas9 cutting mediated by a Pol II promoter-driven sgRNA was evident, it was less effective than that by a U6-driven sgRNA (current Fig. 2a and original Fig. S2b). As the reviewer has pointed out, such result is an indication that Pol II-mediated expression of sgRNA is less abundant compared to a Pol III system, which is consistent with previous publications^{5, 6}. The cutting experiments are now described in more detail, and previous references regarding the Pol II-driven sgRNA efficiencies are added (line 151-157).

We agree with the reviewer that lower levels of Pol II promoter-mediated sgOFF expression may limit its effectiveness. Therefore, to enhance the adaptability of a CRISPRa/i to the inhibitory action by sgOFF, the output sgRNAs (sgTGTa/i) were also introduced in a Pol II-format (instead of a more efficient Pol III construct) in all related experiments (current Fig. 2, 3 and Supplementary Fig. 2, 3). Such reasoning is added to the Discussion (line 511-513).

At the current stage, as the reviewer has rightfully suggested, an off-switching sgRNA (especially one driven by a Pol II promoter) would be likely to fall short of forming a tight control against a CRISPRa/i actuator. We also made efforts in the revision to report the incomplete effects by the sgOFF NOT gate, alongside the documentation of results. Practically, the non-optimal performance by the inhibitory sgRNA indeed fueled our continued explorations which led to the later establishment of a much improved NOT gate (current Fig. 4-6).

We additionally thank the reviewer for pointing out the appeared high background in the CRISPR cutting reporter (original Fig. S2b, sgCon), which might have resulted from a subpar experiment. We also appreciate the comment about the gating on mRuby. The results in the original manuscript were from analyses of total cells, in which we previously failed to set up for mRuby gating. We therefore performed repetition experiments to determine the fluorescence levels of mRuby and EGFP (current Fig. 2a). Both CMV- and U6-sgRNA led to increases of EGFP expression in mRuby⁺ cells (compared with their corresponding controls). Low levels of background EGFP positivity were observed mostly in cells with high mRuby expression, consistent with the data in a previous report⁷. We speculate that the background is likely to be attributed to certain degrees of stop codon readthrough.

Figure S2c

How does sgTGTi behave when expressed from a U6 promoter? This control is required

to establish the robustness of the system. The construct 5xISRE dCas9-KRAB appears to be leaky. A mild expression of Fla-dC-KRAB is detectable in basal conditions and, while the IFN α treatment induces a substantial upregulation of the target gene, the basal expression levels are, in my opinion, enough to drastically reduce the expression of dEGFP (please compare eGFP lanes with IFN α 0 in SgCon-i and sgTGT-i transfections). This design leads to basal dEGFP downregulation regardless the IFN α input, due to leaky dCas9 expression.

[Authors' reply]

We thank the reviewer for these comments and suggestions. First, we performed additional experiments to test the effects by U6- and CMV-driven sgTGTi under parallel conditions, together with 5xISRE-dCas9-KRAB (the original CMV-TGTi-only data has now been replaced). The results showed that although IFN-induced inhibition of EGFP via U6-sgTGTi appeared more prominent, the effect by CMV-sgTGTi was also evident (current Supplementary Fig. 2b and see quantitation). We also added experiments by rendering dCas9 expression constitutive (CMV), and establishing a circuit with an inducible 5xISRE-sgTGTi (current Supplementary Fig. 2c). With this alternative design, IFN treatment also notably down-regulated EGFP. Therefore, Pol II-driven sgRNA expression system represents a functional tool to direct CRISPRi activities, despite featuring a reduced robustness compared to a conventional Pol III system. These results are now detailed (line 158-177).

As the reviewer has pointed out in our original data, in the added experiments using ISRE-dCas9-KRAB, the basal CRISPRi activities in the absence of IFN were likewise observed with both U6- and CMV-TGTi (current Supplementary Fig. 2b, arrows in the quantitation panel). Moreover, in cells transfected with CMV-dCas9-KRAB and ISRE-sgTGTi, the basal suppression of EGFP reporter was also apparent (current Supplementary Fig. 2c, the arrow in the quantitation panel). In this series of experiments, given the constitutive expression of either sgRNA or dCas9-KRAB, such background CRISPRi activities reflected the basal expression of their corresponding partners even under a signal-induced promoter. We reason that such background activity could be reduced by placing both dCas9 and sgRNA under signal-dependent promoters. Next, most of our ensuing experiments in Fig. 2, 3 were carried out with such “co-regulation” design. This construction strategy is described in the Results (line 179-181).

However, we agree with the reviewer that background CRISPRa/i activities may persist in promoter-controlled systems, causing potential complications. We now duly underscore the issues with basal CRISPRa/i activities in the Discussion (line 494-504). Indeed, we believe that these issues further highlight the advantages in our overall design (AND-NOT logic) to improve the gene circuits' targeting specificity. Here, the inhibitory module in the confirmatory NOT gate would act to suppress the basal CRISPRa activation in non-targeted cells. Such comments are added also to the Discussion (line 504-507).

Figure 2b

The leaky expression is once again interfering with the overall design logic,

undermining the final outcome. Basal dCas9 and (consequently) eGFP expression can be appreciated in IFN α treatment (western blot and IFN- cells).

Technical comment: Scalebar and type of experiment of the figure inset are not described.

[Authors' reply]

We thank the reviewer for finding other indications of CRISPRa/i background activities in our data. As mentioned in response to the previous inquiry, further discussions regarding the basal CRISPRa/i activities are added in the Discussion (line 494-504).

We also apologize for the lack of clarity in the figure legends. The required information is now added (line 1059-1060).

Figure S2d

Given the residual expression of dCas9-KRAB in sgISRE-i +IFN α , the lack of reduction of eGFP is somewhat surprising (while similar levels are enough to induce eGFP downregulation in FigS2c). It seems that the sgOFF control is tighter on the sgTGTi and this negative regulation might be sufficient to exert a similar effect. Given that PolIII sgRNA are suboptimally expressed as deduced by FigS2b, they might represent the rate-limiting factor of the gate logic, rather than the dCas9-KRAB module. Can the authors comment on this and add experimental conditions (controls) in which the P1 promoter on the dCas9-KRAB is constitutively combined with 5xISRE sgTGTi?

[Authors' reply]

We thank the reviewer for closely examining our results and for suggesting additional experiments to clarify the contributions from sgOFF's inhibition on either sgTGT and/or dCas9-KRAB, to an overall regulation of CRISPRi function.

As per reviewer's suggestion, we performed additional experiments in a parallel system containing a constitutive CMV-dCas9-KRAB and an inhibitable 5xISRE-sgTGTi. We named this particular system "1xOFF" to indicate a single regulatory point by an sgOFF. In contrast, the one shown in the original data was here named as "2xOFF", corresponding to the double targeting of both dCas9-KRAB and sgTGTi by an sgOFF. We now also emphasize that the "sgOFF" used throughout the manuscript is a generalized nomenclature (line 196-197). In these experiments, a sequence derived from the ISRE promoter was selected as the target for an sgOFF (specifically referred to as "sgISRE" in the original manuscript). It is currently listed as "sg#1-i" to keep a consistent naming pattern for different sgOFFs (line 198-200).

The experiments were carried out under the same condition as before. Notably, with the 1xOFF system, the U6-sgISRE only partially reversed IFN-induced suppression of EGFP (current Supplementary Fig. 2e, left), unlike the apparently complete effect seen in the 2xOFF system (original Fig. S2d and current Supplementary Fig. 2e, right). Although the residual dCas9-KRAB levels associated with the 2xOFF system (as noted by the reviewer) does suggest that the inhibition on sgTGTi may have a greater role, the comparisons between the 1xOFF and 2xOFF systems demonstrate the need to reduce the expression of both dCas9-KRAB and sgTGTi for a more effective blockade of CRISPRi function. A detailed account of such comparative analyses is now

integrated in the Results (line 198-215).

Furthermore, in the ensuing experiments involving sgOFF, we continued with such a design of multiplexed inhibition (see the original and current Fig. 2c-f and Fig. 3).

Figure S2e

It is not clear if the sgOFF sequence is the same of FigS2d or it's a different one. Again, upon target optimization, it seems that, although it exerts its function, sg#2-i is not as efficient as its U6 driven counterpart (FigS2d). A mild downregulation of eGFP can still be appreciated, and dCas downregulation is not as tight as in FigS2d (in which was already not complete). On the other hand sg#1-i is seemingly not working. Despite the elegant logic, it seems to me that, in view of the final application of the system, the polII dampened sgRNA expression efficiency must be resolved.

[Authors' reply]

We thank the reviewer for these comments. Different from the earlier results (original Fig. S2d, current Supplementary Fig. 2e) where the U6-driven sgOFF was utilized, the experiments here explored the effect by an sgOFF expressed under a less robust Pol II system. It was noted that under the CMV promoter, the sgISREi became much less effective in suppressing ISRE-dCas9-KRAB levels (data now shown). Besides the reduced expression of sgISREi under a Pol II promoter, we reasoned that its targeting site or sequence might also not be optimal, which might be masked in an efficient Pol III system. For the purpose of potentially enhancing sgOFF-mediated the inhibitory effects, we engineered the dCas9-KRAB and sgTGTi constructs by inserting multiple target sites near their transcriptional starts (described in the original Fig. S2e legends). The “sg#1-i” or “sg#2-i” (the original version) each matched a different inserted sequence in the corresponding constructs. Here, only one of the two sgOFFs could notably reduce the dCas9-KRAB expression and partially restored the output of EGFP (original Fig. S2e, **current Supplementary Fig. 2f**). These results demonstrate the possibility of target site designing/screening to improve the efficiency of the Pol II-sgOFF system. We now provide more details to such line of efforts (line 216-229). In addition, a softer tone was used to describe the effects by the sg#2-i.

In addition, we apologize for having not clearly explained our nomenclature system. We used “sgOFF” as a general term, and specified each different sgOFF with various names (originally as “sgISREi”, “sg#1-i”, “sg#2-i”, etc.). As mentioned above, we now add clarification for the term “sgOFF” (line 196-197). For further simplification, we now specify all sgOFFs using numbers in the names. Therefore, the original sgISREi is the current sg#1-i. In accordance, the original sg#1-i, sg#2-i, sg#3-i become the current sg#2-i, sg#3-i and sg#5-i, respectively. On the other hand, the name for the most extensively used sg#4-i stay unchanged. The changes in the names for sgOFFs have also been updated in the **current Supplementary Table 3** containing sequences for sgRNA constructs.

Figure s2f

The unique nomenclature of the sgRNA OFF is again confusing, as it appears to be referred to another sequence if I understood properly. Can the author comment on why they didn't insist on using sgISRE-i in Figure S2d?

[Authors' reply]

We sincerely apologize for such clarity issues. As mentioned above, we now add clarification for the term “sgOFF” (line 196-197). Also as mentioned above, the lack of inhibitory efficiency by sgISRE-i under a Pol II promoter, was the reason for searching other inhibitory sgRNAs (line 216-219). Here, the “sg#3-i” and “sg#4-i” (the original version) correspond to unique sequences in the 5xISRE-dCas9 and -sgRNA promoters (either covering the junction of ISRE/miniCMV, or that of miniCMV/multiple cloning site). Such details are now added to the legend (current Supplementary Fig. 2g, a better-quality repetition is now presented). Note that while the name for sg#4-i stayed unchanged, the original sg#3-i is currently named as sg#5-i, as mentioned above.

Figure 2c

Compared to the experiment shown in FigS2f, with the same sgRNAs, the introduction of the MS2-p65-HSF1 module under the control of the 5xISRE element, appears to significantly enhance dCas9 leaky expression compared to a constitutively expressed cassette (FigS2f). This is counterintuitive as it is suggestive of crosstalk between sgOFF (which should be deprived of MS2 stem-loops) and the MS2-p65-HSF1 module. Additionally, leaky expression in absence of INFa is not completely abrogated.

[Authors' reply]

We thank the reviewer for pointing out the high background dCas9 expression in this experiment. Here, the high background levels of dCas9 in the original result do not represent a consistent trend. Additionally, as the reviewer has rightfully reasoned, no crosstalk should exist between a normally-scaffolded sgOFF and the MPH. The original data have been replaced with a piece of more representative result (current Fig. 2c). Practically, certain basal expression/activity of CRISPRa/i is likely to be present even when their components are under signal-dependent promoters. Although such background activities may be reduced by a co-expressed sgOFF, a complete background abrogation is unlikely to be achieved due to the modest potency for such an off-switch. As mentioned above, we particularly add a segment in the Discussion to comment on the background CRISPRa/i activities (line 494-507). Besides, the limitations for the sgOFF system are also now elaborated in both the Results and Discussions (line 288-298, 515-522).

Figure 2e, f

Although the system is working, leaky expression of Cas9 and sgTGTA sgRNA contribute to the AND/NOT logic to not be completely bullet-proof. In a real-life scenario, the system would induce leaky expression of target genes in normal cells too.

[Authors' reply]

We agree with the reviewer on the limitation of the sgOFF strategy. We modified the corresponding text to ensure a factual tone in our description (line 244-247). Again, the limitations for the sgOFF system are also now elaborated in both the Results and

Discussions (line 288-298, 515-522).

Figure S4a

Western blot on mRuby is not a direct proof of increased expression levels of ACRmax compared to AcrIIA4. Also, the superior efficiency of ACRmax is not strongly supported by additional data (e.g. similar eGFP induction in Fig S4b). In absence of additional data, the choice of ACRmax is somewhat arbitrary, although the authors should not stress its superior efficiency based on codon optimization and mRuby western blot only. A western blot of eGFP levels in Figure S4b experiment could be performed to clarify this point.

[Authors' reply]

We thank the reviewer for pointing out some lack of clarity in our data. We concur that the levels of downstream IRES-mRuby, although being a correlative marker, are not formal indications for levels of AcrIIA4 variants with differential codon usages. We conducted additional experiments. Cells were transfected with the originally cloned AcrIIA4 (*L. monocytogenes*) or the fully human codon-optimized version (ACRmax), in constructs without the IRES-mRuby. As per reviewer's suggestion, WB analyses for CRISPRa-driven EGFP levels were used as readout (current Fig. 4a, line 306-309). Although the AcrIIA4 in bacterial codons showed considerable activities in inhibiting CRISPRa function, the ACRmax was visibly even more effective. For simplicity, the original indirect evidence [using the levels of IRES-mRuby] (original Fig. S4a), as well as a less quantitative EGFP immunofluorescence panel was removed (original Fig. 4b).

1. Santiago-Ortiz, J.L. & Schaffer, D.V. Adeno-associated virus (AAV) vectors in cancer gene therapy. *Journal of Controlled Release* **240**, 287-301 (2016).
2. Limberis, M.P., Vandenberghe, L.H., Zhang, L., Pickles, R.J. & Wilson, J.M. Transduction Efficiencies of Novel AAV Vectors in Mouse Airway Epithelium *In Vivo* and Human Ciliated Airway Epithelium *In Vitro*. *Molecular Therapy* **17**, 294-301 (2009).
3. Goh, A.M. et al. Using targeted transgenic reporter mice to study promoter-specific p53 transcriptional activity. *Proceedings of the National Academy of Sciences of the United States of America* **109**, 1685-1690 (2012).
4. Komarova, E.A. et al. Transgenic mice with p53-responsive lacZ: p53 activity varies dramatically during normal development and determines radiation and drug sensitivity in vivo. *The EMBO journal* **16**, 1391-1400 (1997).
5. Knapp, D.J.H.F. et al. Decoupling tRNA promoter and processing activities enables specific Pol-II Cas9 guide RNA expression. *Nature communications* **10**, 1490 (2019).
6. Nissim, L., Perli, S.D., Fridkin, A., Perez-Pinera, P. & Lu, T.K. Multiplexed and programmable regulation of gene networks with an integrated RNA and CRISPR/Cas toolkit in human cells. *Molecular cell* **54**, 698-710 (2014).
7. Kim, H. et al. Surrogate reporters for enrichment of cells with nuclease-induced mutations. *Nature Methods* **8**, 941-943 (2011).

Reviewers' Comments:

Reviewer #1:

Remarks to the Author:

From a review of the changes suggested, the authors have implemented all the requested suggestions as well as can be expected, with new in-vivo data demonstrating a increased IFN γ expression and downstream signalling, with an overall reduction in tumour growth which is very promising.

Co-culture experiments appear to work with reasonable efficiency and effectiveness.

Language has been tightened up and they have addresses reviewer twos comments well.

Reviewer #2:

Remarks to the Author:

The authors have excellently addressed my concerns, as well as those of the other reviewer. I feel that the review process has been very constructive, and the paper is now much stronger.